# Atomic structures of respiratory complex III$_2$, complex IV, and supercomplex III$_2$-IV from vascular plants

Maria Maldonado[1], Fei Guo[1,2], James A Letts[1]*

[1]Department of Molecular and Cellular Biology, University of California Davis, Davis, United States; [2]BIOEM Facility, University of California Davis, Davis, United States

**Abstract** Mitochondrial complex III (CIII$_2$) and complex IV (CIV), which can associate into a higher-order supercomplex (SC III$_2$+IV), play key roles in respiration. However, structures of these plant complexes remain unknown. We present atomic models of CIII$_2$, CIV, and SC III$_2$+IV from *Vigna radiata* determined by single-particle cryoEM. The structures reveal plant-specific differences in the MPP domain of CIII$_2$ and define the subunit composition of CIV. Conformational heterogeneity analysis of CIII$_2$ revealed long-range, coordinated movements across the complex, as well as the motion of CIII$_2$'s iron-sulfur head domain. The CIV structure suggests that, in plants, proton translocation does not occur via the H channel. The supercomplex interface differs significantly from that in yeast and bacteria in its interacting subunits, angle of approach and limited interactions in the mitochondrial matrix. These structures challenge long-standing assumptions about the plant complexes and generate new mechanistic hypotheses.

## Introduction

The canonical mitochondrial electron transport chain (mETC), composed of four integral membrane protein complexes (complexes I–IV; CI–CIV) located in the inner mitochondrial membrane (IMM), transfers electrons from NADH and succinate to molecular oxygen. The concomitant pumping of protons (H$^+$) across the IMM establishes an electrochemical proton gradient that is used by ATP synthase to produce ATP (*Nicholls, 2013*). Whereas the atomic details of the respiratory complexes and several supercomplexes (higher-order complex assemblies) are known for yeast, mammals and bacteria (*Vinothkumar et al., 2014*; *Fiedorczuk et al., 2016*; *Gu et al., 2016*; *Letts et al., 2016*; *Wu et al., 2016*; *Zhu et al., 2016*; *Zickermann et al., 2015*; *Blaza et al., 2018*; *Guo et al., 2017*; *Letts et al., 2019*; *Agip et al., 2018*; *Parey et al., 2018*), the high-resolution structural details of the respiratory complexes and supercomplexes of plants have remained mostly unknown.

Complex III (CIII$_2$), also called the cytochrome $bc_1$ complex or ubiquinol-cytochrome $c$ oxidoreductase, is an obligate dimer that transfers electrons from ubiquinol in the IMM (reduced by CI, CII, or alternative NADH dehydrogenases) to soluble cytochrome $c$ in the intermembrane space (IMS) (*Nicholls, 2013*). This redox reaction is coupled to the pumping four H$^+$ to the IMS. CIII$_2$ is composed of three conserved subunits present in all organisms (cytochrome $b$, COB; cytochrome $c_1$, CYC1; and the iron-sulfur 'Rieske' subunit, UCR1), as well as a varying number of accessory subunits present in eukaryotes (*Iwata et al., 1998*; *Xia et al., 2013*; *Xia et al., 1997*). Each CIII monomer contains one low-potential heme $b$ ($b_L$) and one high-potential heme $b$ ($b_H$) in COB, a heme $c$ in CYC1, a 2Fe-2S iron-sulfur cluster in UCR1, as well as two quinone-binding sites (Q$_P$ and Q$_N$ close to the positive/IMS and negative/matrix sides respectively) in COB. Given that CIII$_2$ is a dimer, these sites in each CIII monomer are symmetrical within the dimer in isolation. However, the symmetry

*For correspondence:
jaletts@ucdavis.edu

Competing interests: The authors declare that no competing interests exist.

**eLife digest** Most living things including plants and animals use respiration to release energy from food. Respiration requires the activity of five large protein complexes typically called complex I, II, III, IV and V. Sometimes these complexes combine to form supercomplexes. The complexes are similar across plants, animals and other living things, but there are also many differences.

Detailed structures of the respiratory complexes have been determined for many species of animals, fungi and bacteria, highlighting similarities and differences between organisms, and providing clues as to how respiration works. Yet, there is still a lot to learn about these complexes in plants.

To bridge this gap, Maldonado et al. used a technique called cryo electron microscopy to study the structure of complexes III and IV and the supercomplex they form in the mung bean. This is the first study of the detailed structure of these two complexes in plants. The results showed many similarities to other species, as well as several features that are specific to plants. The way the two complexes interact to form a supercomplex is different than in other species, as are several other, smaller, structural features. Further examination of complex III revealed that it is flexible and that movements are coordinated across the length of the complex. Maldonado et al. speculate that this may allow it to coordinate its role in respiration with its other cellular roles.

Understanding how plant respiratory complexes work could lead to improvements in crop yields or, since respiration is required for survival, result in the development of herbicides that block respiration in plants more effectively and specifically. Further researching the structure of the plant respiratory complexes and supercomplexes could also shed light on how plants adapt to different environments, including how they change to survive global warming.

may be broken when CIII assembles into asymmetrical supercomplexes (*Letts et al., 2016*; *Letts et al., 2019*; *Letts and Sazanov, 2017*). CIII$_2$'s redox and proton pumping occur via the 'Q-cycle' mechanism (*Cramer et al., 2011*), which allows for efficient electron transfer between ubiquinol (a two-electron donor) and cyt $c$ (a one-electron acceptor). To this end, one electron is transferred from ubiquinol in the Q$_P$ site to the 2Fe-2S in the UCR1 head domain. The head domain then undergoes a large conformational swinging motion from its 'proximal' position close to the Q$_P$ site to a 'distal' CYC1 binding site adjacent to heme $c_1$ (*Zhang et al., 1998*). The electron is then transferred *via* heme $c_1$ to cyt $c$ bound to CIII$_2$ in the IMS. Of note, the UCR1 head domain belongs to the opposite CIII protomer relative to COB and CYC1. The second electron donated by ubiquinol is transferred *via* hemes $b_L$ and $b_H$ to a quinone in the Q$_N$ site, reducing it to ubisemiquinone. The cycle is repeated to regenerate ubiquinol in the Q$_N$ site, ultimately reducing two molecules of cyt $c$ and pumping four protons.

In eukaryotes, the large CIII$_2$ accessory subunits exposed to the mitochondrial matrix have homology to mitochondrial processing peptidases (MPP) of the pitrilysin family (*Gakh et al., 2002*). These metalloendopeptidases—composed of an active β subunit and an essential but catalytically inactive α subunit—cleave mitochondrial signal sequences of proteins that are imported into the mitochondria (*Gakh et al., 2002*). Whereas in yeast the CIII$_2$ accessory subunits with MPP homology (ScCor1/Cor2) have completely lost MPP enzymatic activity, the mammalian CIII$_2$ homolog (UQCR1/UQCR2 heterodimer) retains basal activity to only one known substrate (the Rieske subunit) (*Gakh et al., 2002*; *Taylor et al., 2001*). Hence, in yeast and mammals this enzymatic activity is carried out by soluble MPP heterodimers in the mitochondrial matrix. In contrast, in vascular plants, there is no additional soluble MPP enzyme, and all MPP activity is provided by the CIII$_2$ MPP heterodimer (MPP-α/β). Thus, in plants CIII$_2$ serves a dual role as a respiratory enzyme and a peptidase (*Braun et al., 1992*; *Emmermann et al., 1993*; *Eriksson et al., 1994*; *Braun et al., 1995*; *Eriksson et al., 1996*; *Braun and Schmitz, 1995a*; *Glaser and Dessi, 1999*). This integration of respiratory and peptidase activities may have occurred early in eukaryogenesis (*Braun and Schmitz, 1995b*). The bioenergetic implications of this dual function of plant CIII$_2$ remain unknown.

Complex IV (CIV), also called cytochrome $c$ oxidase, transfers electrons from cyt $c$ to molecular oxygen, reducing it to water (*Nicholls, 2013*). The redox reaction is coupled to the pumping of four protons into the IMS. Like CIII$_2$, CIV is composed of three conserved subunits (COX1, COX2, COX3) and a variable number of accessory subunits, depending on the organism. Electrons are transferred from cyt $c$ to oxygen via COX1's dinuclear copper (Cu$_A$), heme $a$ and copper-associated heme $a_3$ (Cu$_B$, binuclear center). The passage of protons from the matrix to the IMS occurs through distinct 'channels' formed by protonatable amino-acid residues (*Rich, 2017*; *Rich and Maréchal, 2013*; *Yoshikawa and Shimada, 2015*; *Wikström et al., 2018*). It is currently believed that, whereas yeast CIV pumps protons through the K and D transfer pathways (named after key amino-acid residues in each pathway), mammalian CIV uses an H channel in addition to the K and D channels (*Rich, 2017*; *Rich and Maréchal, 2013*; *Yoshikawa and Shimada, 2015*; *Wikström et al., 2018*; *Maréchal et al., 2020*). The contribution of the K, D, H pathways in plant CIV has not been characterized.

In the IMM, respiratory complexes can be found as separate entities or as higher-order assemblies known as supercomplexes (*Schägger and Pfeiffer, 2000*). Although it was initially hypothesized that supercomplexes would allow for direct substrate channeling between complexes, evidence has mounted against this view (*Gu et al., 2016*; *Letts et al., 2016*; *Letts et al., 2019*; *Yu et al., 2018*; *Sousa et al., 2016*; *Blaza et al., 2014*; *Fedor and Hirst, 2018*). Instead, supercomplexes may have roles in improving the stability of the complexes, providing kinetic advantages to the electron transfer or reducing the production of reactive oxygen species or of aggregates in the IMM (*Letts and Sazanov, 2017*; *Milenkovic et al., 2017*). Supercomplexes of various stoichiometries between CIII$_2$ and CIV (e.g. SC CIII$_2$+CIV, SC CIII$_2$+CIV$_2$) have been seen (*Schägger and Pfeiffer, 2000*; *Eubel et al., 2004*; *Eubel et al., 2003*). In plants, the CIII$_2$-CIV supercomplex of highest abundance is a single CIII dimer with a single CIV monomer (SC III$_2$+IV) (*Eubel et al., 2004*). High-resolution structures of the model yeast *Saccharomyces cerevisiae* and *Mycobacterium smegmatis* CIII$_2$-CIV supercomplex (SC III$_2$+IV$_2$) have recently been determined (*Hartley et al., 2019*; *Rathore et al., 2019*; *Gong et al., 2018*; *Wiseman et al., 2018*). Although there is currently no high-resolution structure for a mammalian CIII$_2$-CIV supercomplex, the supercomplex between CI, CIII$_2$ and CIV (SC I+III$_2$+IV, the respirasome) shows a distinct interaction interface between CIII$_2$ and CIV relative to the SC III$_2$+IV$_2$ from yeast and bacteria (*Gu et al., 2016*; *Letts et al., 2016*). Similar to that seen in comparative tomographic studies of plant SC I+III$_2$ and bovine and yeast SC I+III$_2$+IV (*Davies et al., 2018*), the above SC III$_2$+IV$_2$ studies revealed that, while the general configuration of the individual CIII$_2$ and CIV are conserved, the location of the binding interface between CIII$_2$ and CIV in the supercomplex is divergent, with different subunits involved in the different organisms. For plant CIII$_2$ and CIV, the only currently available structural information is from low-resolution, 2D-averages of negative-stain EM samples from *A. thaliana* (*Dudkina et al., 2005*) and potato (*Bultema et al., 2009*). High-resolution structures or atomic models for CIII$_2$, CIV or their supercomplexes are not currently available for the plant kingdom.

Here we present the cryoEM structures of CIII$_2$ and SC III$_2$+IV from the vascular plant *Vigna radiata* (mung bean) at nominal resolutions of 3.2 Å and 3.8 Å, respectively. Using focused refinements around CIV, we achieved a nominal resolution for CIV of 3.8 Å. The structures reveal plant CIII$_2$ and CIV's active sites, as well as the plant-specific configuration of several CIII$_2$ and CIV subunits. The structures also show the SC III$_2$+IV binding interface and orientation, which is unique to plants. Additionally, using cryoEM 3D-conformation variability analysis (*Punjani and Fleet, 2020*), we were able to visualize the swinging motion of CIII$_2$'s UCR1 head domain at 5 Å resolution in the absence of substrate or inhibitors. We also observed complex-wide coupled conformational changes in the rest of CIII$_2$. These results question long-standing assumptions, generate new mechanistic hypotheses and provide the initial structural basis for the development of more selective agricultural inhibitors of plant CIII$_2$ and CIV.

## Results

Mitochondria were isolated from etiolated *V. radiata* hypocotyls. The electron transport chain complexes were extracted using the gentle detergent digitonin. The extracted complexes were further stabilized with amphipathic polymers (amphipol) and separated on a sucrose gradient. Given our interest in respiratory complex I, we pooled fractions containing NADH-dehydrogenase activity and

set up cryoEM grids. Details on the sucrose gradient and NADH-dehydrogenase activity of the sample are available in *Maldonado et al., 2020*. Upon 2D classification of the particles in the micrographs, it became evident that the pooled fractions contained not only the complex intermediate CI* (*Maldonado et al., 2020*), but also CIII$_2$ and SC III$_2$+IV. Complex III and CIV subunits were identified in the mixed mitochondrial pooled fraction by mass spectrometry, in addition to CI subunits (*Supplementary file 1a*, see Materials and methods for full dataset availability). Therefore, CIII$_2$ and SC III$_2$+CIV were purified in silico from the micrographs we had previously used to solve the structure of CI* (*Maldonado et al., 2020*). This is an example of 'bottom-up structural proteomics' (*Ho et al., 2020*) on a partially purified sample, in which a single sample was used for the structural determination of multiple respiratory complexes. Processing of the CIII$_2$ and SC III$_2$+IV particles

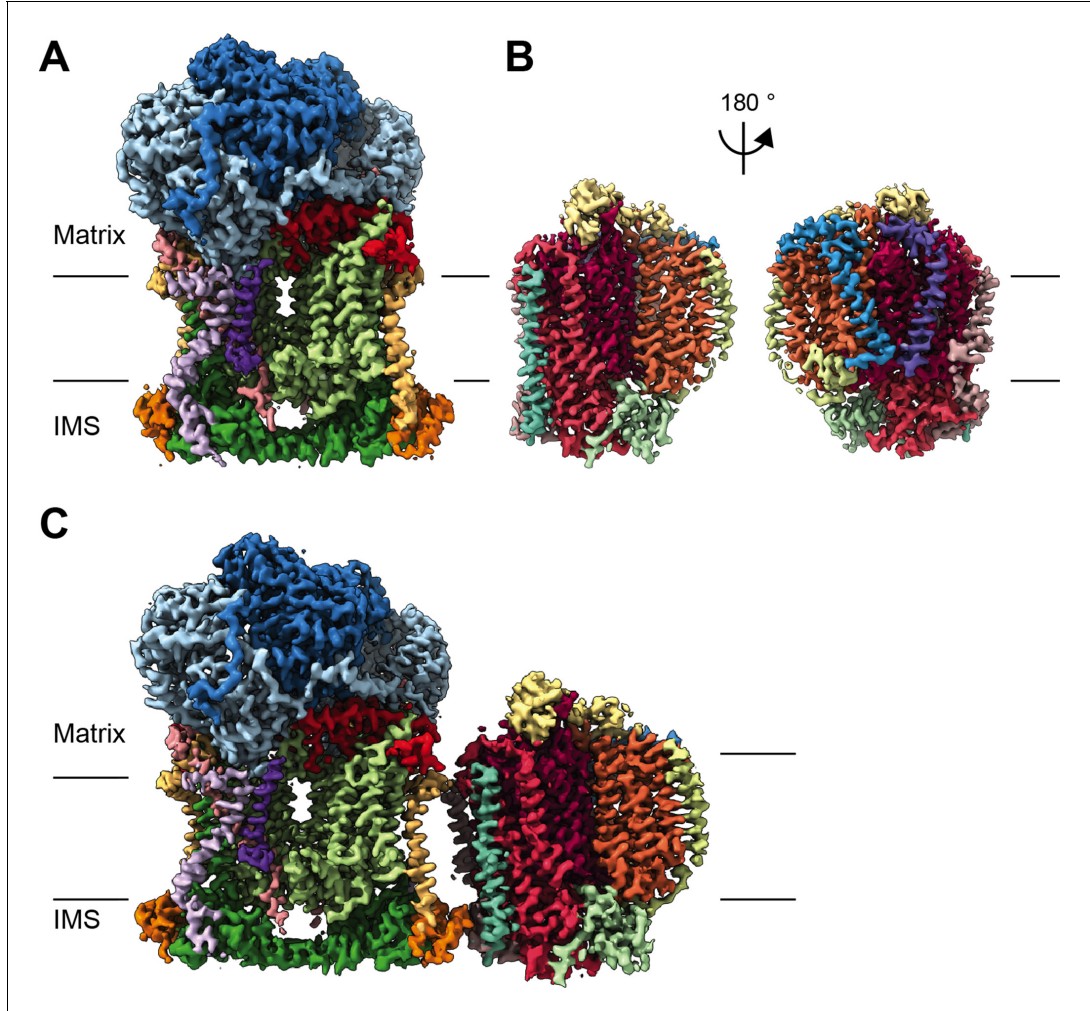

**Figure 1.** CryoEM reconstructions for *V. radiata* mitochondrial CIII$_2$, CIV and SCIII$_2$+IV. (**A**) CryoEM density map for CIII$_2$ in isolation (not assembled into a supercomplex; see also *Figure 1—figure supplement 1* and *Video 1*). (**B**) Density map for CIV, obtained from re-centered focused refinements of CIV in the supercomplex (see also *Figure 1—figure supplement 2* and *Video 2*). (**C**) Composite map of SC III$_2$+IV, assembled by combining CIII$_2$ and CIV-focused refinements from the SC particles (see also *Figure 1—figure supplements 1* and *2* and *Video 3*). Volume surfaces are colored by subunit (see also *Videos 1–3* and *Figures 2* and *5*). The approximate position of the matrix and IMS sides of the membrane are shown with black lines.

The online version of this article includes the following figure supplement(s) for figure 1:

**Figure supplement 1.** Initial processing and reconstructions using cryoSPARC.

**Figure supplement 2.** Supercomplex focused classification and 3D refinement using Relion.

**Figure supplement 3.** Map-Model FSCs are shown for (**A**) CIII$_2$ alone (see also *Figure 1—figure supplement 1*), (**B**) CIV-focused map from the supercomplex particles (see also *Figure 1—figure supplement 2*) and (**C**) the SC III$_2$+IV composite map (see also *Figure 1—figure supplement 2*).

**Figure supplement 4.** Fractionation and activity of extracted mitochondrial membranes.

resulted in near-atomic reconstructions of CIII$_2$ at 3.2 Å, of CIV at 3.8 Å and of SC III$_2$+IV at 3.8 Å (*Figure 1*, *Videos 1–3*, *Figure 1—figure supplements 1–3*, *Table 1*, *Supplementary file 1b*).

Further examination using spectroscopic activity assays (*Belt et al., 2017*) confirmed that pooled fractions of the preparation contained CIII$_2$ respiratory activity (electron transfer from reduced decylubiquinone to cytochrome *c*) that was inhibited by CIII$_2$ inhibitors antimycin A and myxothiazol (*Figure 1—figure supplement 4*). The sample also showed CIV activity from reduced cytochrome *c* to oxygen that was inhibited by CIV inhibitor potassium cyanide (*Figure 1—figure supplement 4*). Although MPP activity of CIII$_2$ was recovered from isolated *V. radiata* mitochondrial membranes (not shown), MPP activity assays (*Braun et al., 1992*; *Teixeira et al., 2015*) of the pooled fractions were inconclusive. Owing to research restrictions during the 2020 COVID-19 pandemic, we were not able to further optimize the peptidase assay for the pooled fractions. However, given the high superposition between the *V. radiata* MPP domain shown here and structures of active MPP (see below), we believe our inability to confirm MPP activity in this case does not significantly impact our interpretation of the data.

## Complex III dimer (CIII$_2$)

### Overall structure and ligands

The *V. radiata* (Vr) structure confirms that VrCIII$_2$ contains 10 subunits per CIII monomer (three conserved, mitochondrially encoded subunits and seven accessory subunits), similar to yeast and mammals (*Figure 2*, *Figure 2—figure supplement 1*). As in mammals and yeast, VrCIII$_2$ contains 13 transmembrane helices per protomer: eight from COB (cyt *b*) and one from each of CYC1 (cyt *c$_1$*), UCR1, QCR8, QCR9, and QCR10 (*Supplementary file 1b*). The MPP domain, composed of two MPP-α/β heterodimers (one per CIII protomer), extends into the matrix. The C-terminus of CYC1 (six α-helices and a 2-strand β-sheet), the entire QCR6 subunit and UCR1's head domain extend into the intermembrane space (IMS). Consistent with their known flexibility and conformational heterogeneity, the linker and head domains of UCR1 (Rieske iron-sulfur subunit) were disordered in our reconstruction and an atomic model was not produced for this region of UCR1. (Throughout the manuscript, we use plant subunit nomenclature unless otherwise stated; see *Supplementary file 1c* for details and name equivalence in other organisms.)

All heme groups (heme $b_H$, $b_L$ and heme $c_1$) were clearly visible and modelled into each cyt *b* and cyt *c$_1$* subunit of the dimer. The distances between the hemes were consistent with that previously seen in other organisms (*Figure 2B and C*; *Cramer et al., 2011*). A catalytic $Zn^{2+}$ ion was also visible and modelled into each MPP-β subunit (*Figure 2B*). Given that we were not able to build an atomic model for the head domain of UCR1, we did not model the iron-sulfur clusters (see 3DVA analysis below for more details).

Density consistent with cardiolipin, phosphatidylethanolamine and phosphatidylcholine lipids was found and modelled into the VrCIII$_2$ map (*Figure 2E*). Similar to that previously seen in yeast, the CIII$_2$ lipids concentrate in lipophilic cavities in COB, CYC1, UCR1, QCR8 and QCR10 close to the $Q_N$ site, both on the surface of CIII$_2$ and at the interface between the CIII protomers. We also observed additional lipids on the exterior of COB close to the $Q_P$ site.

The vast majority of the residues that form the $Q_P$ and $Q_N$ sites and inhibitor-binding pockets in yeast and bovine CIII$_2$ (*Xia et al., 2013*; *Esser et al., 2004*; *Gao et al., 2003*) are conserved in *V. radiata* (*Figure 2—figure supplement 2*). In particular, the COB residues that have been shown to form H-bonds with ubiquinone at the $Q_N$ site of yeast and bovine CIII$_2$ are conserved in *V. radiata* (His208, Ser212, D235), as are the key residues of COB's cd1 helix (Gly149-Ile253) and the PEWY motif (Pro277-Tyr280). Nevertheless, no clear density for

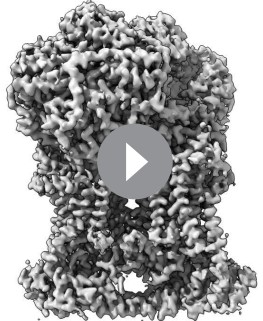

**Video 1.** CryoEM density map and model for *V. radiata* CIII$_2$.
https://elifesciences.org/articles/62047#video1

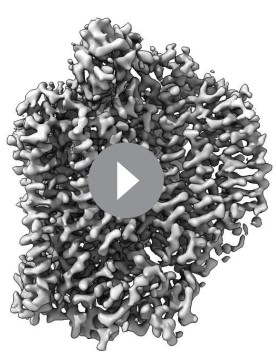

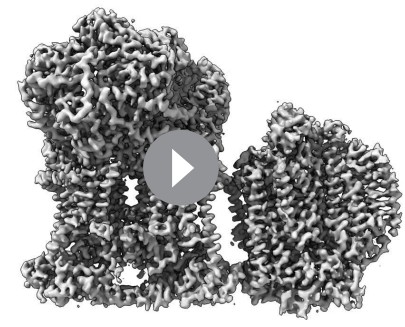

**Video 2.** CryoEM density map and model for *V. radiata* CIV.
https://elifesciences.org/articles/62047#video2

**Video 3.** CryoEM density map and model for *V. radiata* SC III₂+IV.
https://elifesciences.org/articles/62047#video3

binding sites of the *V. radiata* dimer.

endogenous quinone was visible in any of the

It is well established that the mRNAs of several subunits of plant respiratory complexes, including several CI subunits, CIII's COB and CIV's COX1-3, undergo cytidine-to-uridine RNA editing at several sites (*Covello and Gray, 1989*; *Gualberto et al., 1989*; *Hiesel et al., 1989*). This mitochondrial RNA editing is widely conserved across vascular and non-vascular plants (*Takenaka et al., 2013*; *Sper-Whitis et al., 1996*), and most generally acts to restore consensus conserved sequences (*Small et al., 2020*). Inspection of the cryoEM density for the VrCOB subunits showed that the *A. thaliana* COB RNA-editing sites (*Bentolila et al., 2008*; *Giegé and Brennicke, 1999*) are conserved in *V. radiata*. Moreover, *V. radiata* COB contains additional RNA edit sites present in wheat, potato and rice (*Ito et al., 1996*; *Zanlungo et al., 1993*; *Supplementary file 1d*). Given the conservation pattern and the fact that the cryoEM density at these sites was unambiguous, residues were mutated to the edited residues in our model. RNA editing was also identified in *V. radiata*'s COX1 and 3 (see CIV section), as well as in CI subunits (*Maldonado et al., 2020*).

## Differences in mitochondrially encoded subunits and non-MPP accessory subunits

Although the sequence conservation of the *V. radiata* (Vr) mitochondrially encoded CIII₂ subunits (COB, CYC1, UCR1) with respect to *Saccharomyces cerevisiae* (Sc) and bovine (*Bos taurus*, Bt) homologs is modest (~50%), their structural conservation is high (0.75–0.95 Å main chain RMSD, *Figure 2—figure supplements 2–3*). For UCR1, we are only able to compare the regions for which we built an atomic model, that is the N-terminal loop and the main helix but not the head domain. Whereas the helix is highly structurally conserved, VrUCR1 shows an extended N-terminal unstructured loop and short helix that provide more extensive contacts with MPP-β (see MPP section below).

Several of the accessory subunits of VrCIII₂ show significant differences with their yeast and bovine homologs. VrMPP-α and -β, which show the most notable differences, are described in detail in the following section. Here, we discuss VrQCR7 and VrQCR8 (*Figure 2—figure supplement 3H–J*). As in mammals and yeast, VrQCR7 (ScQcr7p, BtUQCRB) contains a helix that contacts the MPP-β anchoring β-sheet (*via* VrCYC1 and VrQCR8), a helix that contacts MPP-β in the opposite CIII monomer and a helix that contacts COB's surface on the matrix side (COB's BC, DE and FG loops, and helices G and H; see *Figure 2—figure supplement 2* for COB helix nomenclature). Nevertheless, similar to the bovine homolog, VrQCR7 is missing an N-terminal helix that in yeast provides several additional contacts with cyt *b*'s helix H and FG loop (*Figure 2—figure supplement 3H*). Similar to mammals and yeast, VrQCR8 (ScQcr8, BtUQCRQ) provides one strand to the multi-subunit β-sheet that helps anchor the MPP domain to the rest of CIII₂ (*Figure 2—figure supplement 3I and J*). Additionally, VrQCR8's transmembrane helix contacts VrCOB's helices G and H. Nevertheless, like BtUQCRQ but in contrast to ScQcr8, VrQCR8 lacks a short perpendicular helix that stacks below cyt

**Table 1.** Cryo-EM data collection, reconstruction, model refinement and validation statistics.

**Data Collection and processing**

| | | | | | |
|---|---|---|---|---|---|
| Microscope | Titan krios (UCSF) | | | | |
| Camera | K3 detector equipped with GIF | | | | |
| Magnification | 60,010 | | | | |
| Voltage (kV) | 300 kV | | | | |
| Electron exposure (e$^-$/Å$^2$) | 51 | | | | |
| Defocus range (μm) | −0.5 to −2.0 | | | | |
| Pixel size (Å) | 0.8332 | | | | |
| Software | SerialEM | | | | |
| Reconstruction | CIII$_2$ | SCIII$_2$+IV | CIII$_2$ focused | CIV-focused | SC Composite |
| Software | cryoSPARC | cryoSPARC | Relion | Relion | Phenix |
| Number of particles | 48,111 | 28,020 | 38,410 | 29,348 | — |
| Box size (pixels) | 512 | 512 | 512 | 512 | 512 |
| Final resolution (Å) | 3.2 | 3.8 | 3.7 | 3.8 | — |
| Map sharpening B factor (Å$^2$) | 67 | 61 | 83 | 77 | — |
| EMDB ID | 22445 | 22449 | 22450 | 22447 | 22448 |

| | | | | |
|---|---|---|---|---|
| Model | CIII$_2$ | | CIV | | SC composite |
| Software | Phenix | | | | |
| Initial model (PDB code) | 6Q9E, 6HU9 | | 6HU9, 5B1A | | 6Q9E, 6HU9, 5B1A |
| Map/model correlation | | | | | |
| Model resolution (Å) | 3.3 | | 3.9 | | 3.9 |
| d99 (Å) | 3.5 | | 3.9 | | 3.9 |
| FSC model 0.5 (Å) | 3.3 | | 3.8 | | 3.8 |
| Map CC (around atoms) | 0.88 | | 0.85 | | 0.84 |
| Model composition | | | | | |
| Non-hydrogen atoms | 32,931 | | 12,772 | | 45,164 |
| Protein residues | 3983 | | 1497 | | 5472 |
| Number of chains | 20 | | 10 | | 30 |
| Number of ligands and cofactors | 8 | | 6 | | 14 |
| Number of lipids | 29 | | 20 | | 43 |
| Atomic Displacement Parameters (ADP) | | | | | |
| Protein average (Å$^2$) | 114.37 | | 38.28 | | 53.17 |
| Ligand average (Å$^2$) | 79.11 | | 50.71 | | 66.25 |
| R.m.s. deviations | | | | | |
| Bond lengths (Å) | 0.005 | | 0.006 | | 0.007 |
| Bond angles (°) | 0.704 | | 0.853 | | 1.107 |
| Ramachandran Plot | | | | | |
| Favored (%) | 93.13 | | 90.75 | | 92.55 |
| Allowed (%) | 6.82 | | 9.18 | | 7.40 |
| Disallowed (%) | 0.05 | | 0.07 | | 0.06 |
| Validation | | | | | |
| MolProbity score | 1.97 | | 2.19 | | 2.04 |
| Clash score | 10.03 | | 14.21 | | 11.51 |
| Rotamer outliers (%) | 0.03 | | 0.08 | | 0.04 |
| EMRinger score | 2.80 | | 2.32 | | 2.01 |
| PDB ID | 7JRG | | 7JRO | | 7JRP |

*b*'s helix H in yeast (*Figure 2—figure supplement 3I*). Moreover, ScQcr8 also has a longer unstructured N-terminus that extends into the cavity of the MPP domain, contacting helix a and DE loop of COB in the same CIII protomer, as well as the helix a of COB of the opposite protomer. Therefore, the interaction interface between VrQCR8 and VrCOB is reduced relative to that of yeast (*Figure 2—figure supplement 3J*).

## Differences in the MPP domain

Each CIII monomer contains an MPP-$\alpha$/$\beta$ heterodimer (plant MPP-$\alpha$/$\beta$, yeast Cor1/2 and mammalian UQCR1/2). Given that both MPP subunits have concave surfaces that face each other, the MPP-$\alpha$/$\beta$ heterodimer contains a large central cavity. The VrMPP-$\alpha$/$\beta$ dimer shows this overall 'clam shell' configuration, with a highly negative surface in the interior of the cavity (*Figure 3A*, *Figure 3—figure supplement 1A–B*). This negative surface has been shown to interact with the generally positively charged pre-sequences of the MPP substrates (*Taylor et al., 2001*). VrMPP-$\beta$ contains the characteristic inverse Zn-binding HxxEH motif of pitrilysin endopeptidases, as well as all the conserved catalytic residues (*Figure 3—figure supplement 1*; *Gakh et al., 2002*). Our cryoEM map shows density for the $Zn^{2+}$ ion, which is coordinated by residues His137, His141, Glu217 (*Figure 3B*). The fourth $Zn^{2+}$-coordinating atom—the oxygen of the water molecule that exerts the nucleophilic attack on the carbonyl carbon of the peptide bond (*Gakh et al., 2002*)—was not modelled. The glutamate that acts as a general base catalyst of this water molecule (*Gakh et al., 2002*; *Taylor et al., 2001*) (Glu140) is also conserved in *V. radiata*. MPP-$\beta$ residues that were confirmed in the substrate-bound yeast soluble MPP structure (*Taylor et al., 2001*) to be critical for substrate recognition are conserved both in the sequence and structural location in VrMPP-$\beta$ (Glu227, Asp231, Phe144, Asn167, Ala168, Tyr169; *Figure 3B*). Additionally, VrMPP-$\alpha$ contains the flexible glycine-rich stretch believed to be involved in substrate binding and/or product release in soluble ScMPP-$\alpha$ (*Taylor et al., 2001*). Consistent with this conformational flexibility, our cryoEM map shows weak density for VrMPP-$\alpha$ residues 340–344. Despite this structural conservation, the MPP enzymatic activity of our preparation could not be confirmed at this time.

VrMPP-$\alpha$/$\beta$'s sequences are more similar to the yeast (and bovine) soluble MPP subunits than to the respective CIII$_2$ subunits (*Figure 3—figure supplement 1C*). Consistently, the VrMPP-$\alpha$/$\beta$ dimer shows secondary-structure elements that are present in ScMPP-$\alpha$/$\beta$ and BtMPP-$\alpha$/$\beta$ but not in ScCor1/2. For instance, on the posterior surface of the cavity, yeast, bovine and mung bean MPP-$\alpha$ fold into six additional helices compared to ScCor2. Additionally, they contain an extra strand in the $\beta$-sheet underneath this helical bundle (*Figure 3C*, *Figure 3—figure supplement 2*).

Furthermore, some structural features of VrMPP-$\alpha$ and -$\beta$ appear to be specific to plants, compared to the available structures (*Figure 3C–D*, *Figure 3—figure supplements 1–2*, *Videos 4–5*). Firstly, VrMPP-$\alpha$ shows a short two-strand $\beta$-sheet on its posterior surface. Secondly, the extended N-terminus of VrMPP-$\alpha$ wraps over the posterior surface of VrMPP-$\beta$ in the same CIII monomer. Thirdly, a ~ 50 amino-acid N-terminal extension on VrMPP-$\beta$ folds into an alpha helix that forms extensive contacts with MPP-$\alpha$ of the same CIII protomer as well as with VrQCR7 of the opposite protomer. A further difference in the overall configuration of the VrMPP domain comes from VrUCR1, whose longer N-terminus forms extensive plant-specific contacts with MPP-$\beta$'s helical bundle and $\beta$-sheet (*Figure 3—figure supplement 2*).

These additional contacts and secondary-structure elements may serve to stabilize plants' MPP domain. Furthermore, they strengthen the connection between the VrMPP domain and the rest of VrCIII$_2$. These plant-specific features of the MPP domain could play functional roles in plant CIII$_2$'s peptidase activity and, potentially, on the coupling between CIII$_2$'s respiratory and non-respiratory functions (see Discussion).

## Conformational heterogeneity analysis of CIII$_2$

Given CIII$_2$'s known conformational flexibility, for example the essential (*Xiao et al., 2000*; *Obungu et al., 2000*; *Nett et al., 2000*) motion of UCR1's head domain during the Q-cycle, we decided to explore the conformational heterogeneity of our CIII$_2$ particles using cryoSPARC's 3D

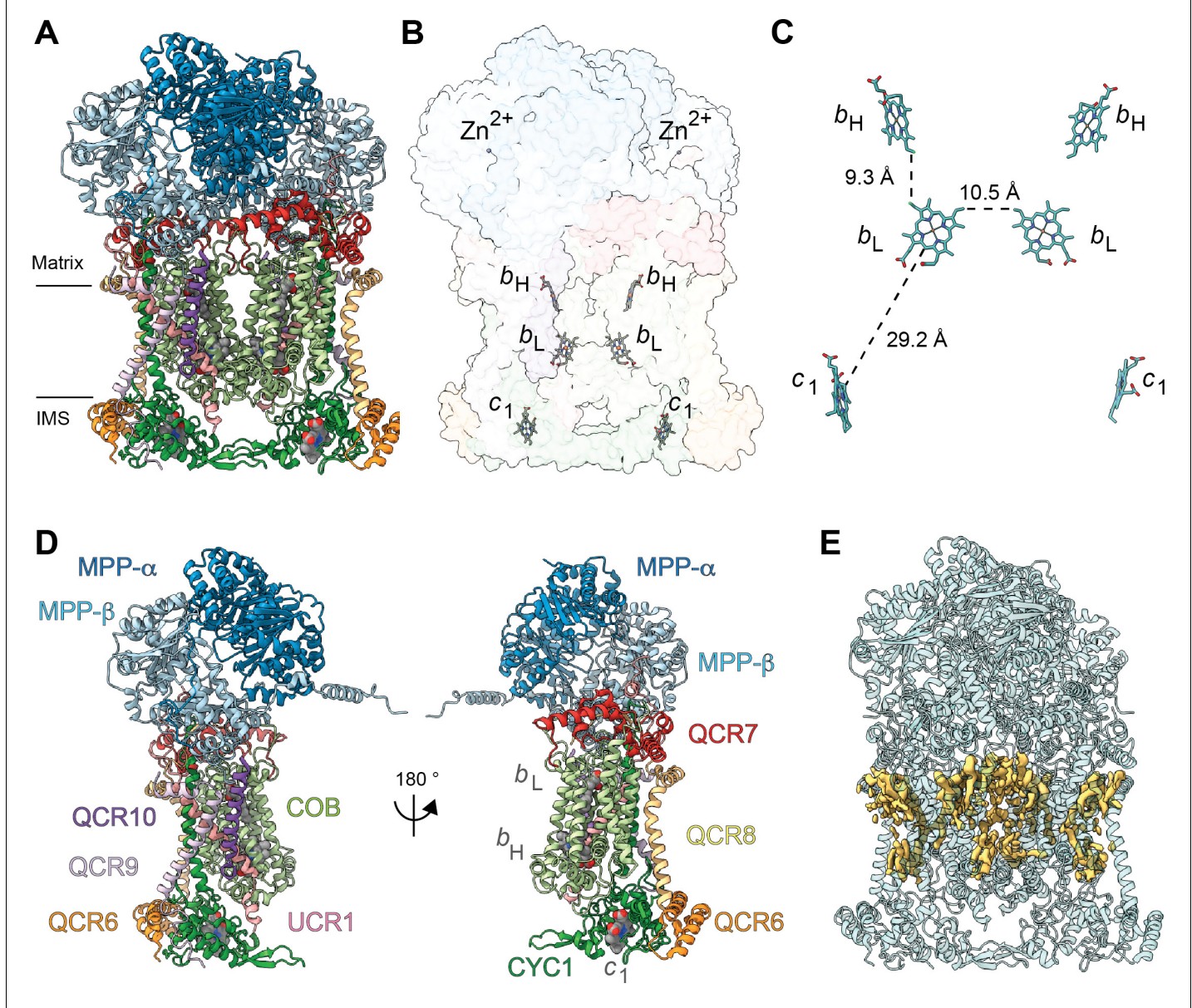

**Figure 2.** Overview of plant $CIII_2$ atomic model. (**A**) $CIII_2$ in cartoon representation with co-factors in sphere representation. The approximate position of the inner mitochondrial membrane is shown with black lines, and the matrix and inter-membrane space (IMS) sides are labeled. (**B**) Position of the observed $CIII_2$ co-factors. Note that the iron-sulfur groups are not shown because the flexible head domain of the iron-sulfur protein is disordered in our cryoEM density. $CIII_2$ shown in transparent surface representation, cofactors in stick representation. (**C**) Distances between the heme groups are shown, calculated edge-to-edge to the macrocyclic conjugated system. (**D**) Each CIII and co-factor are shown as in (**A**) with subunits labeled. The CIII monomers are separated for clarity and the two-fold symmetry axis is indicated. (**E**) Density consistent with lipids (yellow) is shown overlaid on the $CIII_2$ cartoon model (transparent teal). $b_H$, high-potential heme $b$; $b_L$, low potential heme $b$; $c_1$, heme $c_1$; IMS, intermembrane space.

The online version of this article includes the following figure supplement(s) for figure 2:

**Figure supplement 1.** CIII subunit comparison.

**Figure supplement 2.** Comparison of the cyt $b$ conserved subunit (COB) in *V. radiata* (Vr), *S. cerevisiae* (Sc), *Arabidopsis thaliana* (At) and *B. taurus* (Bt).

**Figure supplement 3.** Comparison of select conserved and accessory subunits of $CIII_2$ of *V. radiata* (Vr), *S. cerevisiae* (Sc), *A. thaliana* (At) and *B. taurus* (Bt).

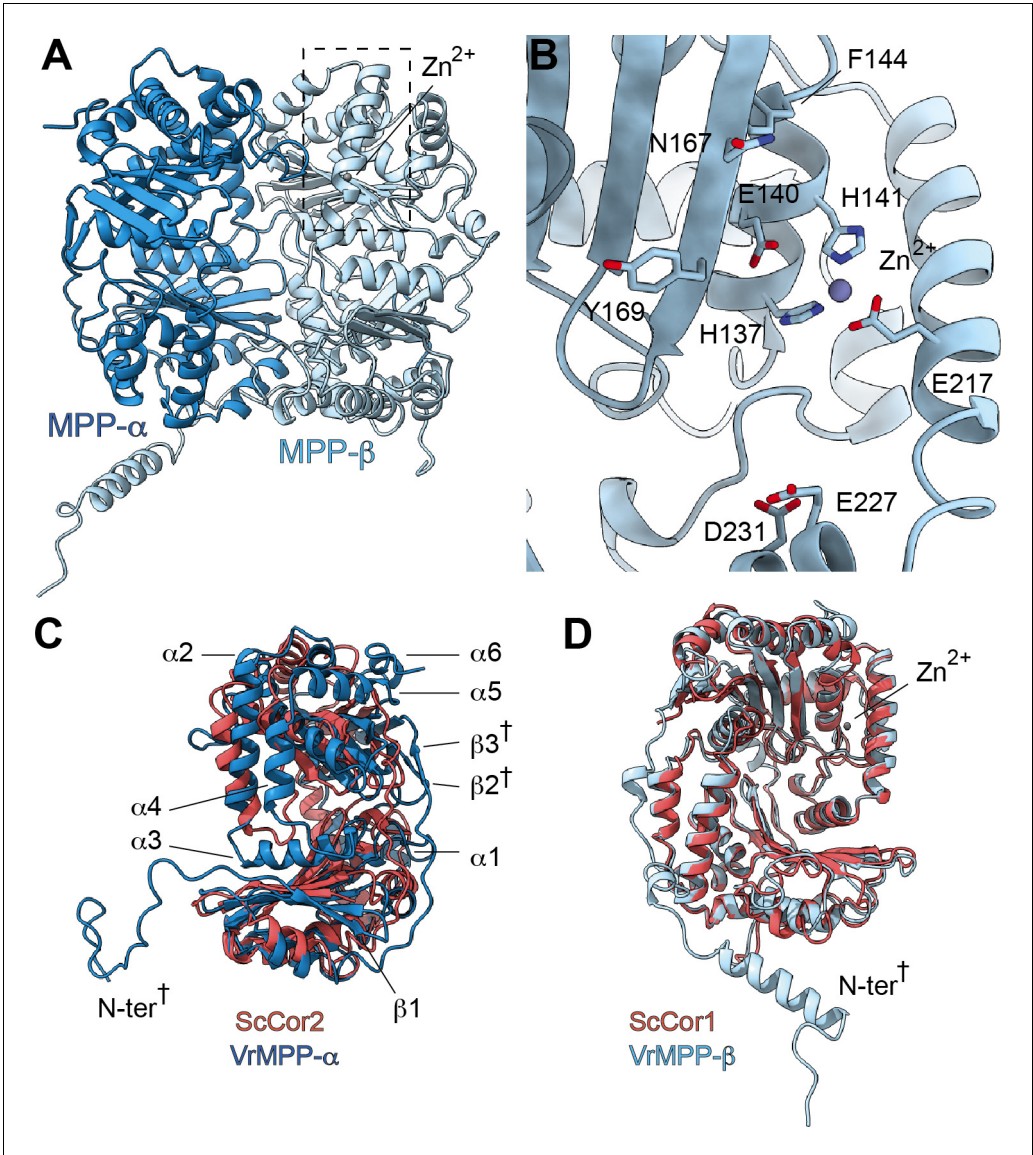

**Figure 3.** *V. radiata*'s CIII₂ mitochondrial processing peptidase (MPP) domain has a conserved architecture and active site but contains plant-specific secondary-structure elements not seen in other CIII-MPP subunits or in soluble MPP. (A) Ribbon representation of the VrMPP-α (blue) and VrMPP-β (light blue) looking into the central cavity. Dashed rectangle indicates the location of the active site, detailed in (B). (B) MPP-β active site [rotated 90° about vertical axis with respect to (A)]. Shown in stick representation are the Zn-coordinating residues (His137, His141, Glu217), the catalytic water-activating residue (Glu140) and conserved, putative substrate-recognition residues (Phe144, Glu227, Asp231, Asn167, Tyr169). Residue Ala168 is also conserved, but not visible in this orientation. (C–D) Superposition of *V. radiata* and *S. cerevisiae* CIII₂ MPP domain subunits. VrMPP-α and -β's structural elements not present in ScCor2 and ScCor1 are marked. Structural elements that are additionally not present in yeast soluble MPP, i.e. plant-specific features, are marked with a cross (†). (D) VrMPP-α (blue) and ScCor2 (dark pink). (D) VrMPP-β (light blue) and ScCor1 (dark pink). β, β—strand; α, α-helix; N-ter, N-terminus. See also *Figure 3—figure supplements 1–2* for further details. *S. cerevisiae* structures from PDB: 6HU9.
The online version of this article includes the following figure supplement(s) for figure 3:

**Figure supplement 1.** Further characterization of VrMPP subunits and their homologs.
**Figure supplement 2.** Alignment of MPP-α homologs.

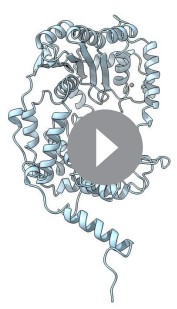

**Video 4.** Superposition of *V. radiata* MPP-β with S. cerevisiae Cor1 (6HU9) and soluble MPP-β (1HR6). https://elifesciences.org/articles/62047#video4

variability analysis (3DVA) (*Punjani and Fleet, 2020*). The 3DVA algorithm uses probabilistic principal component analysis to produce distinct 3D volumes (one per principal component) that reveal the sample's conformational heterogeneity as a continuous motion. Analyzing the individual frames of the motion of the volume, one can reconstruct discrete and continuous conformational changes of the protein sample.

We first analyzed the conformational heterogeneity of the entire CIII₂ low-pass filtered at 6 Å. This allowed for the observation of overall changes at the level of secondary structure. Incorporating data at resolutions higher than 5 Å led to the dominance of high-resolution noise (in particular from the lipid-amphipol belt), precluding clear results. The analysis at 6 Å revealed the lack of QCR10 in some particles, as well as a change in QCR9 (principal components 0–1, *Videos 6–7*), indicating that these subunits only have partial occupancy in the complex. QCR10 was previously found missing in preparations of mammalian CIII₂, likely due to de-lipidation during purification (*Letts et al., 2019*; *Huang et al., 2005*). Given that our purification is very gentle and the supercomplex interactions are maintained, it is unclear whether the changes in QCR10 and QCR9 in a sub-population of particles have biological significance or are simply due to the purification procedure. More importantly, 3DVA revealed that CIII₂ exhibits coordinated 'breathing' motions within and between the protomers of the dimer (components 0–3, *Figure 4A–B*, *Videos 6–9*). The motions, which may be parallel or anti-parallel across the dimer (compare *Videos 6–8* with *Video 9*), extend from the top of matrix-exposed MPP domain to the bottom of the IMS-exposed regions of CYC1, UCR1, and QCR6. This finding contrasts with previous assumptions on CIII₂'s potential for long-range conformational coupling and has implications for the potential interplay between plant CIII₂'s respiratory and peptide-processing functions (see Discussion).

We then examined the conformational heterogeneity of the UCR1 head domain by using a mask around the IMS-exposed domains of CIII₂ at 5 Å. The largest variability component revealed a near-continuous conformational motion in the position of the head domain of UCR1, demonstrating its swinging motion from the proximal *b* position (close to COB's Q_P-site) to its distal *c* position (close to CYC1) during the Q-cycle (*Xia et al., 2013*;

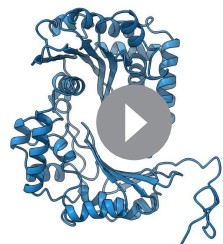

**Video 5.** Superposition of *V. radiata* MPP-α with S. cerevisiae Cor2 (6HU9) and soluble MPP-α (1HR6). https://elifesciences.org/articles/62047#video5

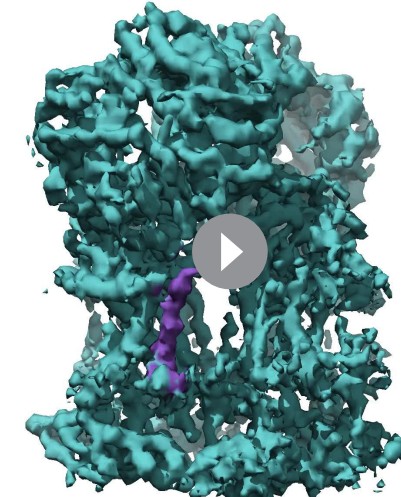

**Video 6.** 3D variability analysis of *V. radiata* CIII₂, component 0. The 3DVA volumes are shown as a continuous movie. CIII₂ in teal, QCR10 in dark purple. https://elifesciences.org/articles/62047#video6

Cooley, 2013; Berry and Huang, 2011; Figure 4C). Moreover, the motions of the UCR1 head domains of the CIII dimer in this variability component were anti-parallel: that is, when one domain is in the proximal position, the other one is in the distal position and vice versa (Figure 4C, Video 10). However, weaker variability components also suggested that the position of UCR1 head domains may also be equivalent in some instances.

Given that the conformational heterogeneity precluded us from building an atomic model of the UCR1 head domain, we rigid-body fit a homology model into the extreme locations of the 3DVA volume and confirmed these corresponded to the Q-cycle proximal and distal positions (Figure 4C, Video 10). At the distal position, the FeS cluster of VrUCR1 was ~11 Å away from CYC1's heme $c_1$ (measured edge-to-edge to the ring system). This is in agreement with previous structures of the UCR1 head domain and within electron-transfer distance (14 Å) (Huang et al., 2005; Palsdottir et al., 2003). Given that we did not model quinone in our

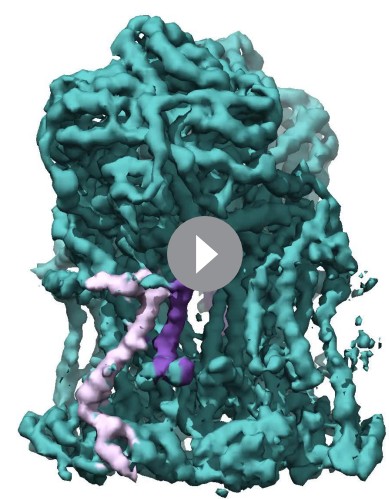

**Video 7.** 3D variability analysis of *V. radiata* CIII$_2$, component 1. The 3DVA volumes are shown as a continuous movie. CIII$_2$ in teal, QCR9 in lilac, QCR10 in dark purple.
https://elifesciences.org/articles/62047#video7

structure, we could not measure its distance to the UCR1 FeS cluster at the proximal position. However, we measured the distance between UCR1's FeS-coordinating residue Cys235 and COB's Tyr285, which is within H-bonding distance to quinone (Palsdottir et al., 2003) (ScRip1-Cys180 and ScCyb-Tyr279 in yeast). This distance (~4.5 Å) was also consistent with those previously seen (Huang et al., 2005; Palsdottir et al., 2003), placing the *V. radiata* FeS cluster within electron-transfer distance to quinone when the head domain is at the proximal position. This confirms that the motion revealed by 3DVA is the expected swinging motion of UCR1. Moreover, given that this flexibility was observed in the absence of substrates or inhibitors, it confirms that UCR1's head domain is intrinsically mobile. Further studies are needed to examine to what extent the movement of the UCR1 head domains are correlated across the CIII dimer and what the mechanistic implications of parallel or anti-parallel movements may be.

## Complex IV
### Overall structure and ligands
The initial resolution of CIV in SC III$_2$+IV was lower than that of CIII$_2$ (Figure 1—figure supplement 1). This is due to the fact that during 3D-refinement the particle poses are dominated by the larger CIII$_2$ and that there is conformational flexibility at the supercomplex interface. This flexibility between CIII$_2$ and CIV within SC III$_2$+IV has been previously seen in cryoEM reconstructions of CIII$_2$-CIV supercomplexes in bacteria and yeast (Hartley et al., 2019; Rathore et al., 2019; Gong et al., 2018; Wiseman et al., 2018). Nonetheless, focused refinements around CIV resulted in an improved reconstruction with a nominal resolution of 3.8 Å for CIV, which allowed for atomic model building (Figure 5, Figure 1—figure supplement 2, Video 2).

At this resolution, all prosthetic groups (heme *a*, heme $a_3$, dinuclear copper A and copper B at the binuclear center), as well as a Mg$^{2+}$ ion and a Zn$^{2+}$ ion were visible and modelled into the structure (Figure 5B–C). The residues that coordinate the prosthetic groups, including the covalent bond between Cε-Tyr247 and the Nε-His243 that coordinates copper B in VrCOX1, are clearly visible and hence conserved in *V. radiata*. The distances between the prosthetic groups are consistent with those seen in other organisms (Figure 5C; Rich, 2017). Additionally, density consistent with cardiolipin, phosphatidylethanolamines and phosphatidylcholines was seen and modelled into the CIV map (Figure 5D). These lipids and acyl chains are located in hydrophobic cavities of VrCOX1 and

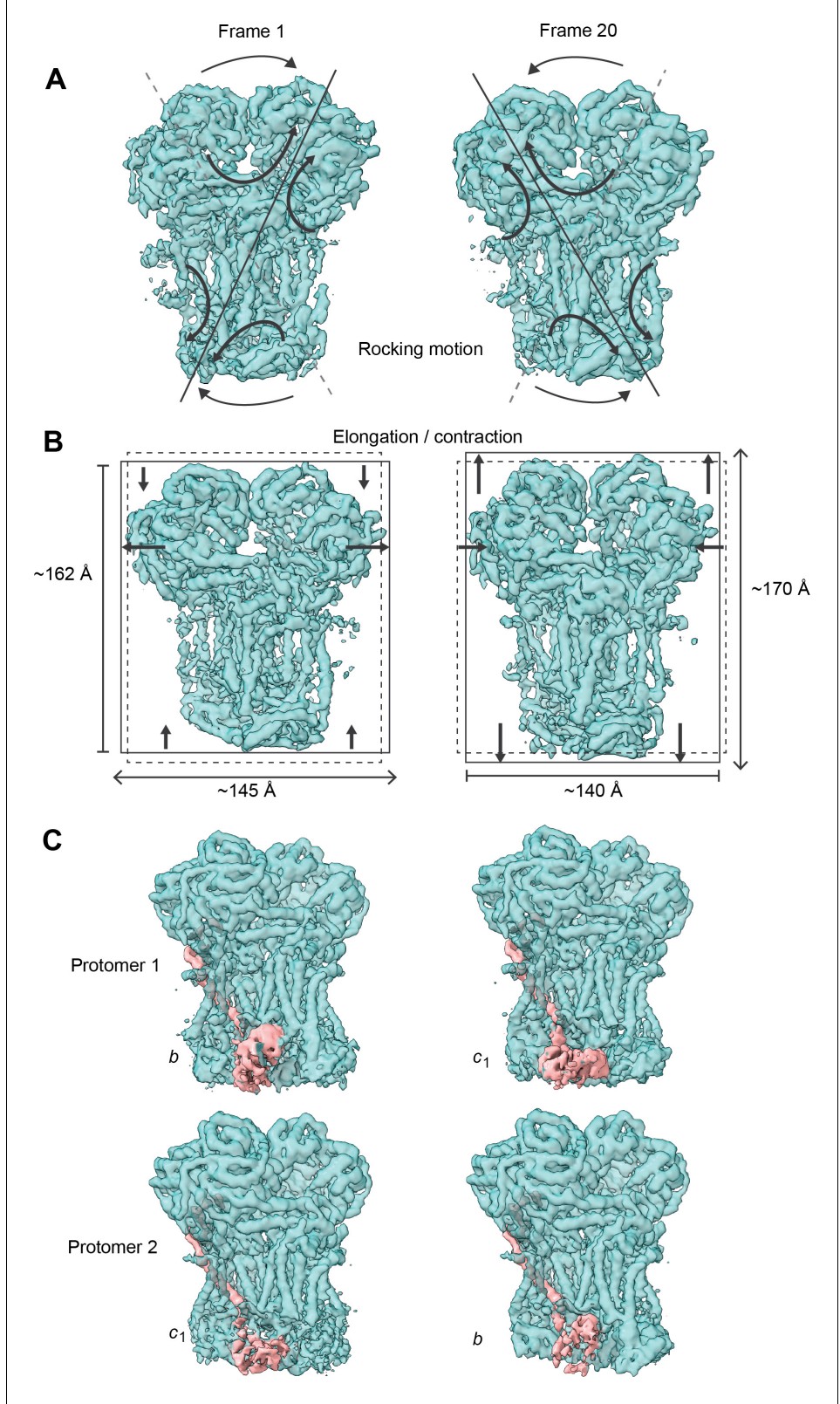

**Figure 4.** Conformational heterogeneity analysis of *V. radiata* CIII$_2$ reveals complex-wide, coordinated movements and shows the swinging motion of UCR1 in the absence of substrates or inhibitors. (A–B) CIII$_2$-wide motions revealed by principal component 2 (A) and 3 (B). Frame 1 (left) and frame 20 (right) of the continuous motion of CIII$_2$ (teal surface) are shown. Black arrows indicate the motion. (A) Rocking motion of CIII$_2$. Solid lines indicate the main axis of the rocking. Dashed lines indicate the axis of the other frame for comparison. (B) Elongation and contraction of CIII$_2$ in the vertical and

*Figure 4 continued on next page*

*Figure 4 continued*

horizontal directions. Solid rectangles indicate the edges of the complex in that frame. Dashed rectangles indicate the edges of the complex in the other frame, for comparison. The dimensions of the rectangle sides are given in Å. (C) Frame 1 (left) and frame 20 (right) of the continuous motion of the UCR1 head domain shown for $CIII_2$ protomer 1 (top) and protomer 2 (bottom). The density corresponding to UCR1 is shown in pink. The position of the UCR1 head domain is indicated by $b$ (proximal) or $c_1$ (distal). Note that, when protomer one is in the $b$ position, protomer two is in the $c_1$ position and vice versa. See *Videos 6–9* for the motion of all components.

VrCOX3, in similar locations to those seen in yeast and bovine CIV. Moreover, conserved RNA-editing sites were unambiguously identified in VrCOX1 and VrCOX3 and thus changed in the model (*Supplementary file 1d*).

## Plant CIV composition

The subunit composition of plant CIV has remained unclear, with different mass spectrometry studies suggesting up to 13 subunits, including some putative plant-specific subunits (*Millar et al., 2004*; *Klodmann et al., 2011*; *Senkler et al., 2017*; *Elina Welchen and Mansilla, 2018*). Our structure shows that VrCIV is composed of 10 subunits (three conserved, mitochondrially encoded subunits and seven accessory subunits). In contrast, mammalian CIV contains 11 accessory subunits and yeast CIV contains 9. Thus, although VrCIV's overall architecture is similar to other organisms', given the lack of certain homologs (ScCox26, ScCox6/BtCOX5a, BtCOX7b, BtCOX8, BtNDUFA4), there are significant differences (*Figure 6*, *Figure 6—figure supplement 1*).

The presence of the *V. radiata* CIV subunits in the structure was confirmed by mass spectrometric analysis of our cryoEM sample (*Supplementary file 1a*). Our structure confirms the identity of plant COX-X2 as the homolog of mammalian COX4 (ScCox5), COX-X3 of mammalian COX7c (ScCox8) and COX-X4 of mammalian COX7a (ScCox7) (*Millar et al., 2004*). The putative plant-specific subunits COX-X5, GLN2, ABHD18, AARE and PRPK (*Millar et al., 2004*; *Klodmann and Braun, 2011*) were not observed in our mass spectrometry sample or our structure (*Supplementary file 1a*). Although two peptides of the putative subunit COX-X1 were identified in our mass spectrometry sample, the lack of unassigned density in our cryoEM reconstruction large enough to constitute an additional subunit demonstrates that this protein is not present in SC $III_2$+IV of etiolated *V. radiata* tissues. The possibility remains that free CIV and/or CIV in supercomplexes of non-etiolated tissues may have a different subunit composition.

## Differences in conserved and accessory subunits

Among the VrCIV subunits that have mammalian and yeast homologs, structural conservation is generally high, particularly for the mitochondrially encoded subunits (*Figure 5—figure supplement 1*). The conserved covalent bond between $N\varepsilon$-His243 and $C\varepsilon$-Tyr247 on the HPEVY ring of VrCOX1 is clearly seen in our density (*Figure 5—figure supplement 1J*). Moreover, VrCOX2 contains the highly conserved residues Tyr255, Met362 (BtCOX2-Tyr105, Met207) believed to be part of the electron transport path from cyt $c$ to $CU_A$ of CIV (*Shimada et al., 2017*).

A significant difference is seen in VrCOX4, which is missing a ~100 amino-acid N-terminal helical domain compared to its homologs (ScCox5, BtCOX4; *Figure 6A and C*). In yeast and mammals, this N-terminal helical domain mainly interacts with ScCox6/BtCOX5a, which is one of the accessory subunits that is missing in plants (*Supplementary file 1c*). Moreover, in

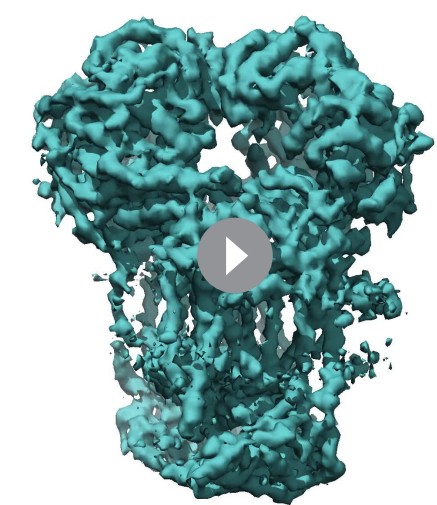

**Video 8.** 3D variability analysis of *V. radiata* $CIII_2$, component 2. The 3DVA volumes are shown as a continuous movie. $CIII_2$ in teal.
https://elifesciences.org/articles/62047#video8

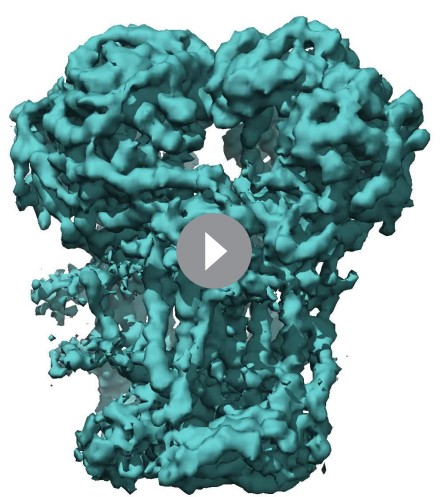

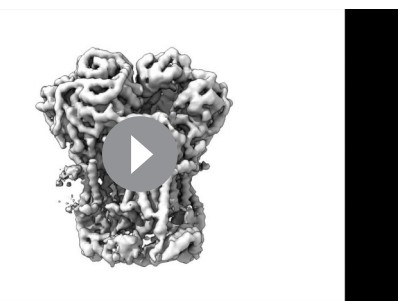

**Video 10.** Swinging motion of the *V. radiata* UCR1 head domains. The 3DVA volumes are shown as a continuous movie. A *V. radiata* UCR1 head-domain homology model was rigid-body fit into the 3DVA volume.
https://elifesciences.org/articles/62047#video10

**Video 9.** 3D variability analysis of *V. radiata* CIII₂, component 3. The 3DVA volumes are shown as a continuous movie. CIII₂ in teal.
https://elifesciences.org/articles/62047#video9

yeast SC III$_2$+IV$_2$, this helical domain of ScCox6 interacts with MPP-β of CIII$_2$ to provide the main CIII:CIV contacts for supercomplex formation. Hence, this interface is absent in the plant SC III$_2$+IV (see SC section below).

Further differences are found in VrCOX4's C-terminus, which shows an extended loop that reaches towards the IMS side of VrCOX2 and then folds back towards the solvent-accessible face of CIV. Another difference in the vicinity of VrCOX4 is seen in VrCOX5c, which lacks the C-terminal helix that in BtCOX6c makes additional contacts to BtCOX2 and BtCOX4 (*Figure 6A*, *Figure 6—figure supplement 1*). These differences in VrCOX4 and VrCOX5c—and the lack of a homolog for ScCox5/BtCOX4—are notable, as these subunits form the majority of interactions between CIV and CIII$_2$ in the plant SC III$_2$+IV (see SC section below). Differences in VrCOX5b (shorter N-terminus), VrCOX7a (longer N-terminus with additional contacts to VrCOX3) and VrCOX7c (different path for the unstructured N-terminus) compared to their mammalian counterparts (*Figure 6—figure supplement 1*) are also likely related to the fact that these subunits provide SC-interface contacts in mammals not observed in plants. However, whether these differences are a cause or an effect of the divergent supercomplex binding interfaces remains to be determined.

Additional differences are seen in VrCOX6a (ScCox13, BtCOX6a). In this case, the plant subunit is more similar to the bovine homolog. Its shorter helix interacts with VrCOX1 and VrCOX3 on the IMS side but does not provide matrix-side interactions as the yeast homolog does (*Figure 6—figure supplement 1*).

## CIV's proton transfer pathways

Translocation of protons through CIV in different organisms occurs via the D, K, and H 'channels' (proton transfer pathways) of the COX1 subunit (*Rich, 2017*; *Rich and Maréchal, 2013*; *Yoshikawa and Shimada, 2015*; *Wikström et al., 2018*). The D and K channels are essential for coupled proton transfer in CIV of bacteria, yeast and mammals (*Rich, 2017*; *Rich and Maréchal, 2013*; *Yoshikawa and Shimada, 2015*; *Wikström et al., 2018*; *Maréchal et al., 2020*). However, whereas mutations of key H channel residues abolish proton pumping in bovine CIV (*Shimokata et al., 2007*; *Tsukihara et al., 2003*), analogous mutations have no effect on growth, respiratory rate or proton-to-electron ratios of yeast CIV (*Maréchal et al., 2020*). Thus, it is currently thought that the H channel only plays a role in proton transfer in mammalian CIV (although it may still act as a dielectric channel in yeast [*Rich and Maréchal, 2013*; *Maréchal et al., 2020*]). The contribution of the D, K, and H channels to CIV proton transfer in plants is unknown.

Sequence and structural analyses show that all protonatable residues of the D and K channels of yeast and mammals are conserved in VrCOX1 (*Figure 7*, *Figure 7—figure supplement 1*). In contrast, several of the residues of the mammalian H channel are not conserved in mung bean

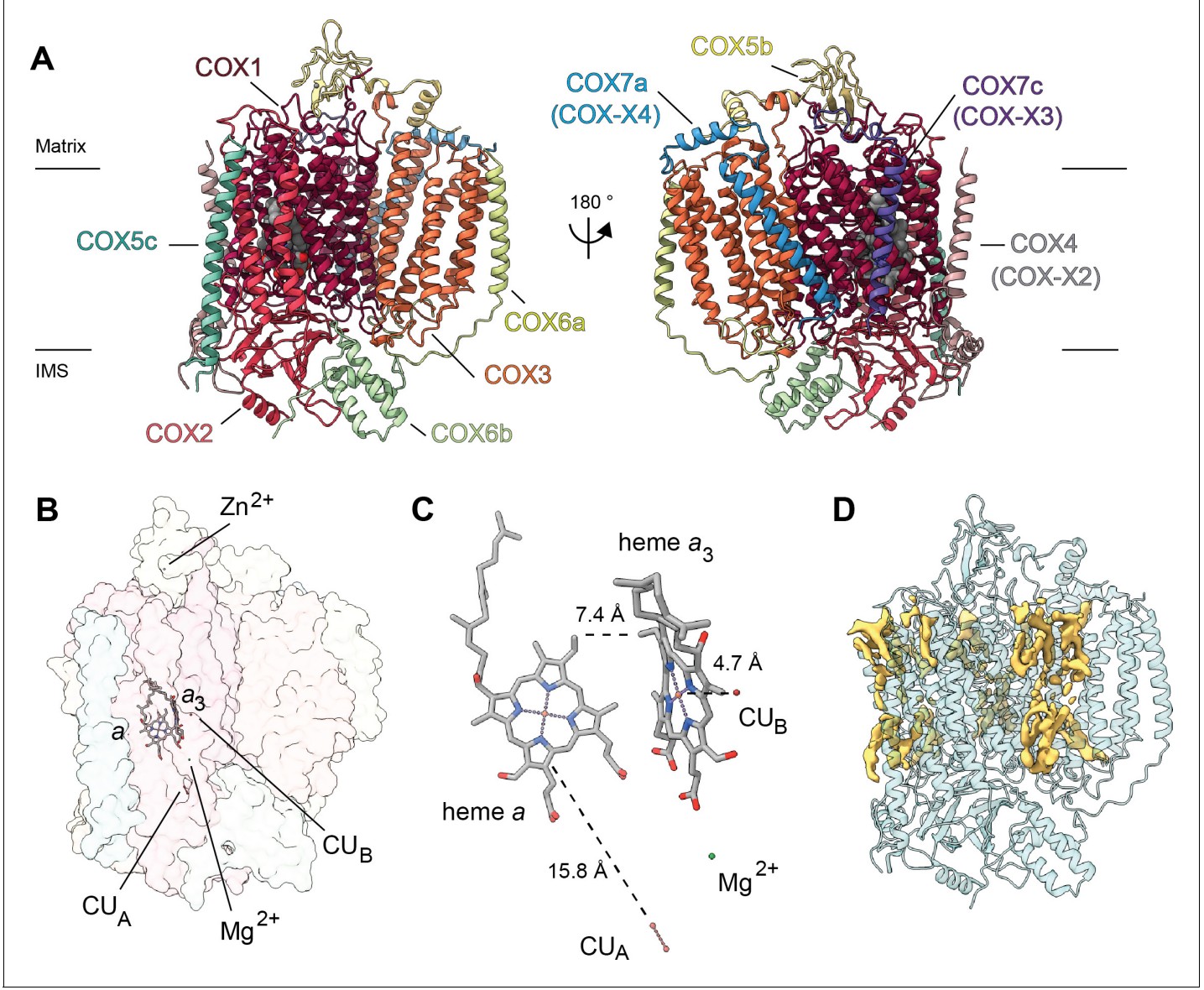

**Figure 5.** Overview of *V. radiata* CIV and its co-factors. (A) CIV in cartoon representation colored by subunit with co-factors in sphere representation colored by atom. The position of the inner mitochondrial membrane is indicated with black lines, and matrix and inter-membrane space (IMS) are labeled. Names of the subunits previously believed to be plant-specific are given in parentheses. (B) Position of the observed CIV co-factors. CIV shown in transparent surface representation, cofactors in stick representation. (C) Edge-to-edge distances between the heme groups and the copper co-factors are shown. The co-factors are rotated 20 degrees relative to (B) for clarity. (E) Density consistent with lipids (yellow) is shown overlaid on the CIV cartoon model (transparent teal). *a*, heme a; $a_3$, heme $a_3$.

The online version of this article includes the following figure supplement(s) for figure 5:

**Figure supplement 1.** Comparison of the COX1-3 subunits in *V. radiata* (Vr), *S. cerevisiae* (Sc), *A. thaliana* (At) and *B. taurus* (Bt).

(*Figure 7—figure supplement 1*). In bovine's H channel, the amide bond between Tyr440 and Ser441 and an H-bond network towards Asp51 are essential features for proton translocation (*Yoshikawa and Shimada, 2015*; *Shimokata et al., 2007*; *Tsukihara et al., 1996*). None of these key bovine residues are conserved in *V. radiata* (or *S. cerevisiae*) (*Figure 7*, *Figure 7—figure supplement 1*). Moreover, in the bovine H channel mutation of Ser441 to Pro441 abolishes proton pumping (*Shimokata et al., 2007*). The corresponding amino-acid for BtCOX1-Ser441 in both *V. radiata* and yeast is proline (VrCOX1-Pro443; ScCox1-Pro441). Additional amino-acid differences between *V. radiata* and *B. taurus* are seen at the entrance and exit of the H channel

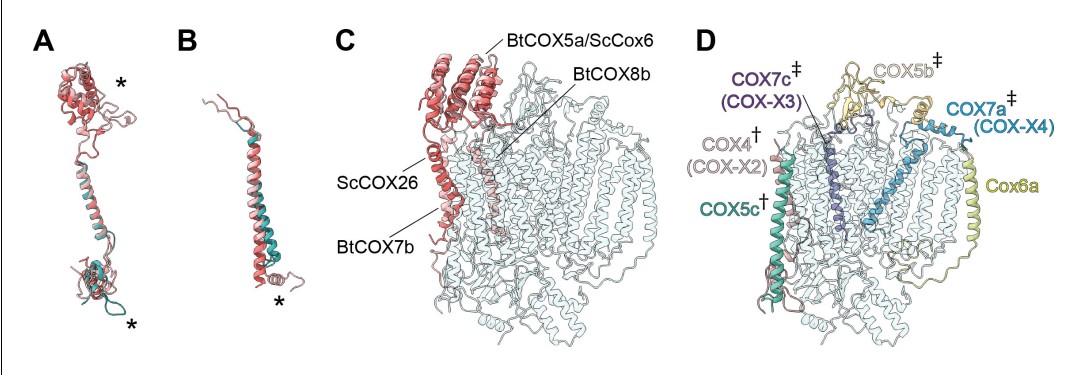

**Figure 6.** Subunit differences in *V. radiata* CIV. (**A–C**) Superposition of subunits of *V. radiata* (Vr, teal), *S. cerevisiae* (Sc, dark pink; PDB: 6HU9) and *B. taurus* (Bt, light pink; PDB: 5B1A) CIV. Subunits were aligned with the corresponding *V. radiata* subunit. Asterisks mark the differences discussed in the text. Names of the subunits previously believed to be plant-specific are given in parentheses. (**A**) Superposition of VrCOX4 (COX-X2), ScCox5a, BtCOX4. (**B**) Superposition of VrCOX5c, ScCox9, BtCOX6c. (**C**) Superposition of the yeast and bovine CIV subunits that do not have homologs in *V. radiata*, onto the *V. radiata* CIV model (transparent teal). Alignment by COX1 subunits. (**D**) Location of the *V. radiata* accessory subunits that show notable differences with their yeast/bovine homologs. Subunits that form the supercomplex interface in *V. radiata* are marked with (†). Subunits whose homologs form the supercomplex interface in the *B. taurus* respirasome are marked with (‡).

The online version of this article includes the following figure supplement(s) for figure 6:

**Figure supplement 1.** CIV subunit comparison.

---

(*Figure 7*). The above suggests that, similar to yeast, the H channel is not a proton transfer pathway in plant CIV and that proton transfer in plant CIV occurs *via* the D and K channels. Experimental evidence is needed to confirm this hypothesis.

## Supercomplex III$_2$-IV (SC III$_2$+IV)

By docking the individually refined CIII$_2$ and CIV models into the SC III$_2$+IV composite map (*Figure 1*, *Figure 1—figure supplement 2*, *Video 3*), we were able to define the binding interface of the plant supercomplex. Direct contacts between the complexes occur in one site in the matrix side and one site in the IMS (*Figure 8*). Site 1 (matrix side), shows a single hydrophobic contact between one residue of VrQCR8 (Pro31) and one residue of VrCOX2 (Trp59) (*Figure 8B*). Our cryoEM reconstruction also contains a short stretch of weak, unassigned density near the first modelled residue of VrCOX5c (BtCOX6c, ScCox9), which could maximally represent an additional four amino acids. If so, the N-terminus of VrCOX5c could potentially provide additional contacts with CIII$_2$. However, this density may also correspond to a bridging lipid bound between the two complexes.

The limited matrix-side contacts in *V. radiata* are in stark contrast to the supercomplex interface in *S. cerevisiae*, where binding is dominated by interactions between ScCor1 (VrMPP-β) and ScCOX5a (VrCox4) on the matrix side. Instead, *V. radiata*'s site 2 (IMS side) provides the bulk of the protein-protein interactions of the supercomplex, with a hydrophobic interaction between VrQCR6 and VrCOX5c (Leu26 and Leu51 respectively) as well as an interface between VrQCR6 (Pro19 and Lys20) and VrCOX4 (Arg114-Phe117) (*Figure 8C*). Despite the potential for lipid bridges at the matrix leaflet of the IMM, there are no direct contacts in the membrane and no protein contact at the IMS leaflet of the IMM. The overall limited binding interactions between *V. radiata*'s CIII$_2$ and CIV, and, consequently, the lower expected stability of the plant supercomplex compared to yeast's, are consistent with the fact that (CIII$_2$+CIV)$_n$ supercomplexes have only been experimentally identified in a few of the plant species studied (*Dudkina et al., 2006*; *Braun, 2020*).

Unsurprisingly given the large differences in contacts, there is a significant difference in the angle between CIII$_2$ and CIV in *V. radiata versus* yeast, resulting in a more 'open' orientation in *V. radiata* (*Figure 8—figure supplement 1*). This orientation leads to a difference of 18° in the angle between the CIII$_2$ and CIV as measured by the relative positions of the $b_h$-hemes in CIII$_2$ and the $a$-hemes in CIV. The difference in orientation also results in a larger estimated distance between the CIII$_2$- and CIV-bound cyt $c$ in *V. radiata* (~70 Å) than in yeast (~61 Å) (*Figure 8—figure supplement 1E*).

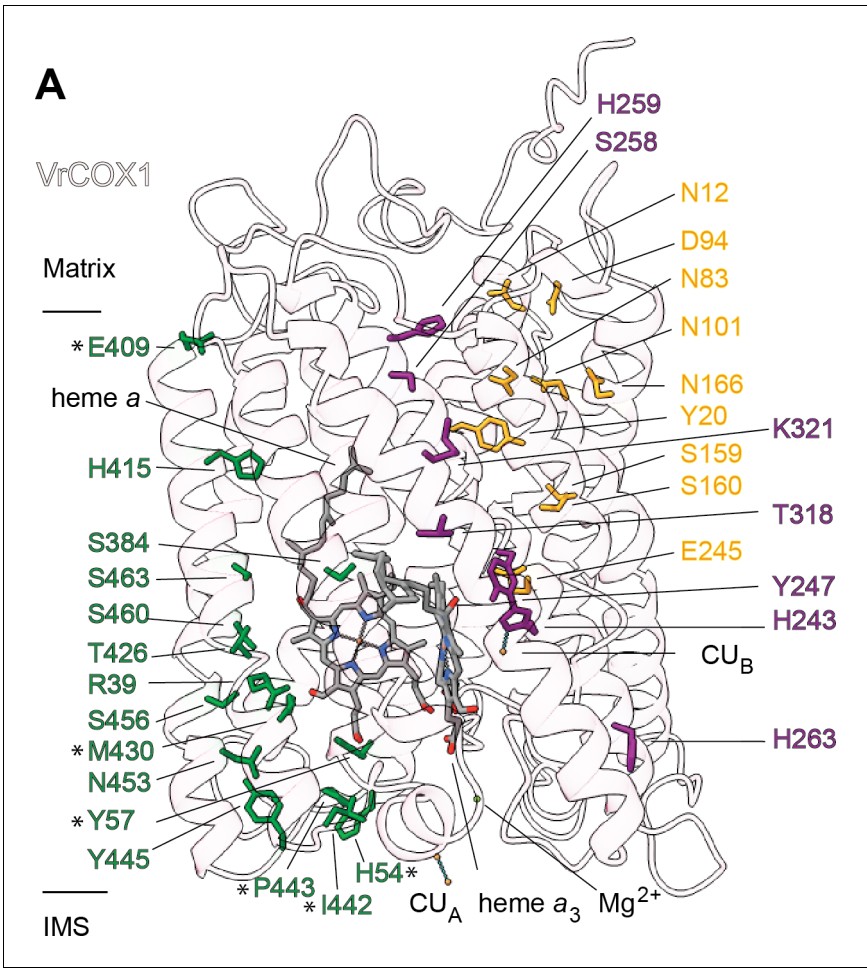

**Figure 7.** Proton transfer pathways of *V. radiata* CIV. VrCOX1 (transparent ribbon), co-factors (stick) and key residues (colored stick) are shown for the D channel (yellow), K channel (purple) and H channel (green). Proton-channel residues that are mutated in *V. radiata* with respect to *B. taurus* are marked with an asterisk (\*). Approximate position of matrix and IMS ends of the transmembrane region are shown.

The online version of this article includes the following figure supplement(s) for figure 7:

**Figure supplement 1.** Sequence alignment of COX1 highlighting the H, D and K channels.

## Discussion

Here, we present the first structures and atomic models of CIII$_2$, CIV and SC III$_2$+IV from the plant kingdom (*Figure 1*, *Videos 1–3*). These *V. radiata* structures reveal atomic details of the catalytic sites and co-factor binding of plant CIII$_2$ (*Figure 2*) and CIV (*Figure 5*). Moreover, they show plant-specific structural features for several subunits, most notably the MPP subunits of CIII$_2$ (*Figure 3*). Conformational heterogeneity analysis of CIII$_2$ allowed us to observe the swinging motion of UCR1's head domain in the absence of substrates or inhibitors, and revealed coordinated, complex-wide motions for CIII$_2$ (*Figure 4*, *Videos 6–10*). The CIV structure defines the subunit composition of the plant complex and suggests that the proton translocation in plants occur via the D and K, rather than the H channel (*Figures 5*, *6* and *7*). The structures also reveal the plant-specific arrangement of the CIII$_2$-CIV interface in the supercomplex, which occurs mostly on the IMS side of the membrane and at a different angle from that seen in other organisms (*Figure 8* and *Figure 8—figure supplement 1*).

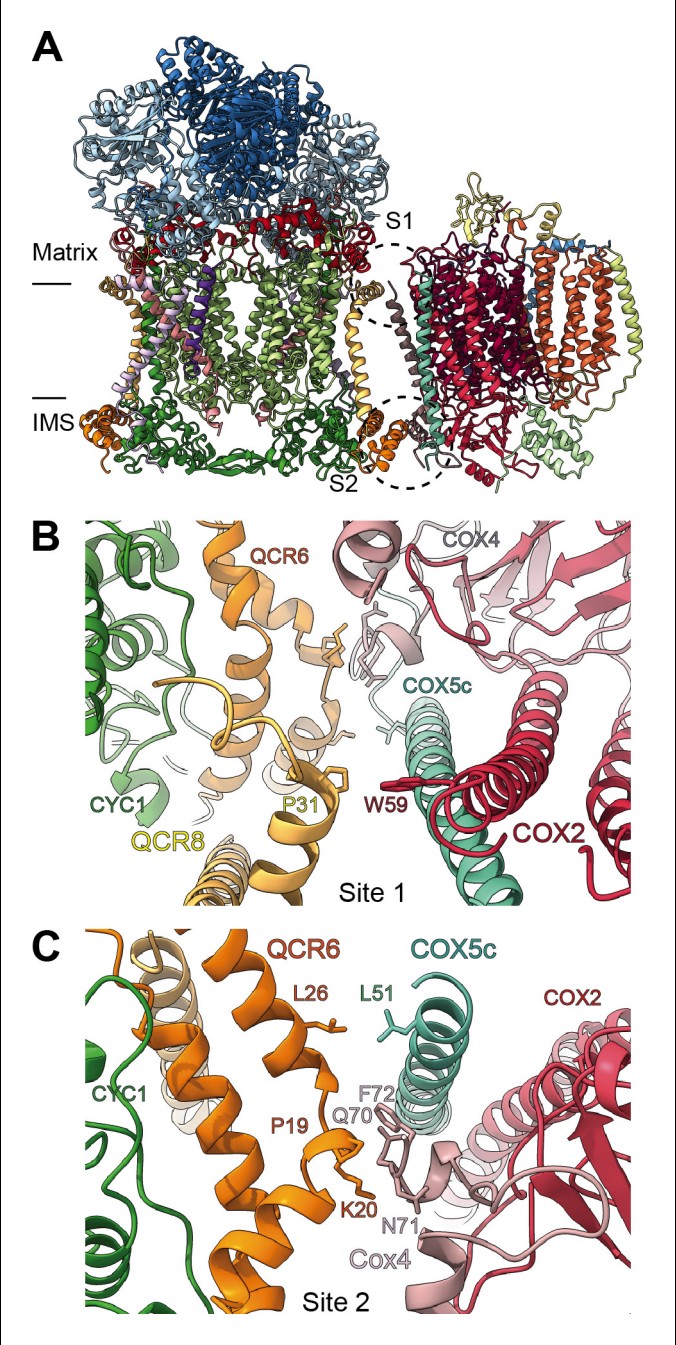

**Figure 8.** SCIII$_2$+IV interface in *V. radiata*. (**A**) General orientation of SC III$_2$+IV in ribbon representation viewed from the membrane. Approximate position of the inner mitochondrial membrane is shown. Sites 1 (S1) and 2 (site 2) of the supercomplex interface are marked in dashed circles. (**B**) Detailed view of the protein-protein interaction in site 1 (Pro31 of VrQCR8 and Trp59 of VrCOX2) with the interacting atoms shown in stick representation. Note that interacting residues of site two appear in stick in the background. (**C**) Detailed view of the protein-protein interaction in site 2 (Pro19-Lys20 of QCR6, Gln70-Phe72 of COX4, Leu26 of QCR6, Leu51 of COX5c) with the interacting atoms shown in stick representation.

The online version of this article includes the following figure supplement(s) for figure 8:

**Figure supplement 1.** Differences in SC III$_2$+IV interactions between *V. radiata* and *S. cerevisiae* (PDB: 6HU9).

## Complex IV subunit composition

Plant CIV subunit composition has been previously analyzed by mass spectrometry of proteins from 2-dimensional blue-native gels (BN-PAGE) (*Millar et al., 2004*; *Klodmann et al., 2011*; *Senkler et al., 2017*). Although these studies were mostly in agreement, several putative CIV subunits—including several putative plant-specific subunits—differed between datasets. Given the considerable technical challenges in obtaining plant CIV samples, experimental evidence for the stoichiometric presence of these putative subunits in complex IV remained limited, with strongest evidence for COX-X1, COX-X2 and COX-X4 (*Senkler et al., 2017*; *Braun, 2020*). The structure of *V. radiata* CIV presented here offers a complementary approach to determine the complex's subunit composition. Our structure shows that CIV obtained from etiolated *V. radiata* sprouts is composed of 10 subunits (three mitochondrially encoded subunits and seven accessory subunits). Only three of the previous plant-specific candidates are seen in our structure (COX-X2, COX-X3 and COX-X4). Moreover, structural analysis shows that COX-X2, COX-X3 and COX-X4 are homologs of mammalian and yeast CIV subunits (BtCOX4/ScCOX5, BtCOX7c/ScCox8 and BtCOX7a/ScCox7, respectively) rather than being plant-specific subunits. Although mass spectrometry analysis of our mixed sample also shows some evidence for the occurrence of COX-X1, this protein is not present in our structure, and its function remains unknown. Our structure provides a new definition for plant CIV's composition; however, we note that the arrangement may differ between free CIV and that in supercomplexes. Moreover, its composition may be dynamically regulated in different metabolic states (e.g. different light or oxygen levels), as is known to occur for certain subunit isoforms in other organisms (*Burke et al., 1997*).

## Conformational heterogeneity of plant CIII$_2$

The 3DVA allowed us to observe the full swing of the UCR1 (Rieske subunit) head domain (*Figure 4*, *Video 10*) without the need for any CIII$_2$ inhibitor to capture the proximal, distal or intermediate positions in the mobile state (*Esser et al., 2019*). As such, it provides direct confirmation for a multitude of previous crystallographic, mutational, kinetic and molecular dynamics studies, mostly done in the presence of inhibitors, that showed the flexibility and mobility of this domain (*Xia et al., 2013*; *Cooley, 2013*; *Berry and Huang, 2011*; *Izrailev et al., 1999*; *Huang and Berry, 2016*). Together, the findings definitively show that the swinging motion is an inherent property of the UCR1 in the absence of substrates. Moreover, the conformational heterogeneity analysis suggests that the movement of the UCR1 head domains is coordinated between the CIII$_2$ protomers to a large degree (*Figure 4C*, *Video 10*). Determining the nature of and mechanism for this inter-protomer coordination will have implications on electron transfer in CIII$_2$ and its supercomplexes.

Unfortunately, however, this conformational flexibility precluded us from building an atomic model for the head domain. Thus, we were not able to evaluate the H-bonding pattern of the UCR1 head domain with COB, or the implications of such pattern to the Q-cycle electron bifurcation mechanism (*Xia et al., 2013*; *Ho et al., 2020*; *Belt et al., 2017*). Similarly, the resolution of the 3DVA was not sufficient to evaluate changes in the positions of COB's cd1 and ef helices. These helices are critical components of the UCR1's COB binding 'crater' (*Xia et al., 2013*), and their position changes in response to binding of different CIII$_2$ inhibitors (*Esser et al., 2006*), with important implications for the Q-cycle mechanism. It is important to note that 3DVA only reveals conformational changes, with no information on kinetics or occupancy rates. Nonetheless, we demonstrate here that cryoEM conformational heterogeneity tools such as 3DVA (*Punjani and Fleet, 2020*) and others (*Zhong et al., 2020*) are a valuable complementary approach to study the conformational changes of CIII$_2$'s Q-cycle in its native state, as well as in the presence of inhibitors and substrates.

Moreover, the 3DVA revealed that CIII$_2$ can undergo different types of complex-wide motions that are coordinated across sides of the membrane and between protomers (*Videos 6–9*). Changes at the 'top' of the complex on one side of the membrane co-vary (i.e. are coupled) with movement at the 'bottom' of the complex on the other side of the membrane. This long-range conformational coupling across the entire CIII$_2$ could be the basis for symmetry-breaking and coordination of the UCR1 head domain motion between the CIII$_2$ protomers.

The long-range conformational coupling is particularly relevant in the context of plant CIII$_2$'s dual roles in signal-peptide processing and respiration. The potential interdependence between these two functions was investigated using CIII$_2$ inhibitors (*Eriksson et al., 1994*; *Eriksson et al., 1996*),

ultimately leading to the interpretation that these functions are independent (*Glaser and Dessi, 1999*). In the presence of the $CIII_2$ respiratory inhibitors antimycin A ($Q_N$-site inhibitor) and myxothiazol ($Q_P$-site inhibitor) at concentrations that inhibit ~90% of spinach $CIII_2$'s respiratory activity, the complex's peptidase activity is inhibited 30–40% (*Eriksson et al., 1994*; *Eriksson et al., 1996*). Given that higher concentrations of inhibitors are needed to elicit MPP inhibition than respiratory inhibition, the authors initially speculated that the effects on the peptidase activity could be due to the inhibitors preventing necessary conformational changes (*Eriksson et al., 1994*). However, when crystal structures of metazoan $CIII_2$ in complex with these inhibitors became available and revealed the locations of the inhibitor binding sites (*Iwata et al., 1998*; *Xia et al., 1997*; *Zhang et al., 1998*), the large distances between these sites and the MPP domain were interpreted to reinforce the notion that the dual roles of plant $CIII_2$ are independent, as long-range coupled conformational changes were deemed unlikely (*Glaser and Dessi, 1999*). In contrast, our 3DVA results showed that long-range coupled motions are intrinsic to *V. radiata* $CIII_2$. Moreover, our atomic model of plant $CIII_2$ revealed additional contacts and secondary-structure elements not previously seen in other organisms that enhance the interaction between the MPP domain and the rest of the complex. For example, the extended N-termini of MPP-$\alpha$ and -$\beta$ bridge across the dimer and provide plant-specific contacts with $CIII_2$'s membrane subunits (*Figure 3*, *Figure 3—figure supplements 1–2*). Moreover, UCR1's longer N-terminus in plants also provides plant-specific contacts with the MPP domain (*Figure 2—figure supplements 1* and *3*; *Figure 3—figure supplement 1C*). Given its span across the membrane and its domain-swapping across protomers, UCR1 may have roles as a 'conformational coupler' beyond its essential function in the Q-cycle.

Together, our 3DVA results challenge long-standing assumptions on plant $CIII_2$'s suitability for conformational coupling and call for a re-evaluation of the relationship between the respiratory and the processing activities of the plant complex.

## Plant supercomplex III$_2$+IV interface

The orientation and binding interfaces of SC $III_2$+IV vary significantly among organisms (*Hartley et al., 2019*; *Rathore et al., 2019*; *Gong et al., 2018*; *Wiseman et al., 2018*; *Sousa and Vonck, 2019*). For instance, the CIII surface used by yeast to bind to CIV is instead used by mammals to bind to CI (*Hartley et al., 2019*). Given these disparities, it is not surprising that the differences seen in the VrCIV subunits are concentrated in the subunits that form the supercomplex interfaces in the different organisms (*Figure 6*). Moreover, while some of the supercomplex interactions in *V. radiata* are reminiscent of the supercomplex interface in yeast, there are significant differences in the protein:protein sites, interacting subunits and angle of orientation within the SC (*Figure 8* and *Figure 8—figure supplement 1*). In yeast, the main interface is on the matrix side, with the N-terminal helical domain of ScCox5a (VrCOX4) interacting with ScCor1 (homolog of VrMPP-$\beta$). In *V. radiata*, this interface is lacking, as plant COX4 does not possess the ~100 amino-acid helical N-terminal domain present in yeast and mammals (*Figure 6* and *Figure 6—figure supplement 1*). In contrast, the main supercomplex interface in mung bean is in the IMS, driven by contacts between VrQCR6 and VrCOX4. In the yeast supercomplex, the homologs of VrQCR6 and VrCOX4 (ScQCR6 and ScCOX5a/b) also interact in the IMS side, but in a much more limited fashion.

In light of VrCIII$_2$'s conformational heterogeneity and the potential interdependence between respiratory and peptidase functions of plant $CIII_2$, an intriguing possibility is that matrix-side interactions between $CIII_2$ and CIV are minimized in the plant supercomplex to prevent steric constraints on the MPP domain, which is catalytically active and likely requires flexibility for its peptidase activity. Thus, the plant-specific supercomplex interface may have evolved to accommodate the particularities of plant $CIII_2$'s dual respiratory and processing functions.

Nevertheless, whereas the details differ, the overall location of the $CIII_2$:CIV interface in *V. radiata* and yeast is similar. A related observation has been made for the supercomplexes between $CIII_2$ and CI (SCI+III$_2$) of plants, yeast and mammals as seen by sub-tomogram averaging (*Davies et al., 2018*). In this case, although the interfaces between CI and $CIII_2$ in the different organisms were also similar, there was a ~10° difference in the angle between CI and $CIII_2$. Additional functional/structural studies of supercomplexes from organisms of diverse phylogenetic origins could determine whether the location of the supercomplex interface has been achieved by convergent or divergent evolution. In turn, this would shed light on the evolution and potential functional significance of the interface sites.

What can already be concluded is that—as seen in yeast (*Hartley et al., 2019*; *Rathore et al., 2019*)—the benefit of the SC III$_2$+IV arrangement in plants does *not* involve direct electron transfer from CIII$_2$ to CIV by simultaneously bound cyt *c* on each complex, as the calculated distance between the bound cyt *c* is too large (~70 Å, *Figure 8—figure supplement 1*). Recent quantitative-proteomics estimations of the stoichiometry of plant respiratory-chain components indicate that the average plant mitochondrion contains ~6500 copies of CIII monomers (i.e. ~3250 CIII$_2$),~2000 copies of CIV and ~2250 copies of cytochrome *c* (*Fuchs et al., 2020*). This implies a maximum of ~2000 copies of SC III$_2$+IV and, thus, a roughly 1:1 ratio between cyt *c* and SC III$_2$+IV. (The ratio has been estimated to be 2–3 in *S. cerevisiae* [*Stuchebrukhov et al., 2020*]). Based on recent theoretical analyses of electron flow between CIII$_2$ and CIV (*Stuchebrukhov et al., 2020*), at this low 1:1 ratio, electron flow would be limited by the time constant of cyt *c* equilibrating with the bulk IMS phase. Under these conditions, the formation of SC III$_2$+IV in the plant mitochondrion would provide a kinetic advantage to electron flow between CIII$_2$ and CIV by reducing the distance between them relative to CIII$_2$ and CIV freely diffusing in the plane of the membrane. It is important to note that this possible kinetic advantage does not imply substrate trapping or channeling between the complexes, and is thus consistent with a single cyt *c* pool (*Stuchebrukhov et al., 2020*).

Our work provides the first high-resolution structure of SC III$_2$+IV in plants, revealing plant-specific features of the complexes and supercomplex. Detailed comparisons of plant CIII$_2$ and CIV sites with existing structures of inhibitor-bound complexes in other species will allow for the development of more selective inhibitors for plant CIII$_2$ and CIV, frequently used as agricultural herbicides and pesticides (*Esser et al., 2014*). The structures also allow for the generation of new mechanistic hypotheses—for example, related to proton translocation in CIV—and a re-evaluation of long-standing assumptions in the field—for instance, related to CIII's capacity for long-range coordinated motion and the relationship between its respiratory and processing functions. Together with biochemical, cellular and genetic studies, further comparative analyses of these atomic structures with the growing number of respiratory complexes and supercomplexes across the tree of life will allow for the derivation of the fundamental principles of the respiratory electron transport chain.

## Materials and methods

**Key resources table**

| Reagent type (species) or resource | Designation | Source or reference | Identifiers | Additional information |
|---|---|---|---|---|
| Biological sample (*Vigna radiata*) | *V. radiata* seeds | Todd's Tactical Group | TS-229 | Lot SMU2-8HR; DOB 2/25/2019 |
| Chemical compound, drug | Digitonin, high purity | EMD Millipore | 300410 | |
| Chemical compound, drug | A8-35 | Anatrace | A835 | |
| Chemical compound, drug | Gamma-cyclodextrin | EMD Millipore | C4892 | |
| Chemical compound, drug | Decylubiquinone | Santa Cruz Biotechnology | sc-358659 | |
| Chemical compound, drug | Equine cytochrome c | Sigma Aldrich | C2506 | |
| Software, algorithm | Clustal Omega | *Madeira et al., 2019* | RRID:SCR_001591 | |
| Software, algorithm | Geneious | *Kearse et al., 2012* | RRID:SCR_010519 | |
| Software, algorithm | SerialEM | University of Colorado, *Schorb et al., 2019* | RRID:SCR_017293 | |
| Software, algorithm | RELION 3.0 | *Zivanov et al., 2018* | RRID:SCR_016274 | |

*Continued on next page*

*Continued*

| Reagent type (species) or resource | Designation | Source or reference | Identifiers | Additional information |
|---|---|---|---|---|
| Software, algorithm | Motioncor2 | *Zheng et al., 2017* | | |
| Software, algorithm | Ctffind4 | *Rohou and Grigorieff, 2015* | RRID:SCR_016732 | |
| Software, algorithm | crYOLO | *Wagner et al., 2019*; *Wagner and Raunser, 2020* | RRID:SCR_016732 | |
| Software, algorithm | Phyre2 | *Kelley et al., 2015* | | |
| Software, algorithm | Coot | *Emsley and Cowtan, 2004* | RRID:SCR_014222 | |
| Software, algorithm | PHENIX | *Liebschner et al., 2019*; *Goddard et al., 2018*; *Pettersen et al., 2004* | RRID:SCR_014224 | |
| Software, algorithm | UCSF Chimera | Resource for Biocomputing, Visualization, and Informatics at the University of California, San Francisco, *Pettersen et al., 2004* | RRID:SCR_004097 | |
| Software, algorithm | UCSF ChimeraX | Resource for Biocomputing, Visualization, and Informatics at the University of California, San Francisco, *Goddard et al., 2018* | RRID:SCR_015872 | |
| Software, algorithm | PyMOL Molecular Graphics System | Schrödinger, LLC | RRID:SCR_000305 | Version 2.0 |
| Software, algorithm | Scaffold Proteome Software | Proteome Software Inc | RRID:SCR_014345 | Version 4.8.4 |
| Software, algorithm | X! Tandem | The GPM | | Version Alanine (2017.2.1.4) |
| Other | Holey carbon grids | Quantifoil | Q310CR1.3 | 1.2/1.3 300 mesh |

## Overall sample and data collection

The cryoEM dataset used in this paper is the same sample, grid and micrographs as those used in *Maldonado et al., 2020*. CryoEM data processing for the structures reported in this paper and those reported in *Maldonado et al., 2020* diverged after 2D classification (see *Figure 1—figure supplement 1*). Further data processing for the structures shown here is described in detail below.

## *Vigna radiata* mitochondria purification

*V. radiata* seeds were purchased from Todd's Tactical Group (Las Vegas, Nevada, USA). Seeds were incubated in 1% (v:v) bleach for 20 min and rinsed until the water achieved neutral pH. Seeds were subsequently imbibed in a 6 mM $CaCl_2$ solution for 20 hr in the dark. The following day, the imbibed seeds were sown in plastic trays on damp cheesecloth layers, at a density of 0.1 $g/cm^2$ and incubated in the dark at 20 °C for 6 days. The resulting etiolated mung beans were manually picked, and the hypocotyls were separated from the roots and cotyledons. The hypocotyls were further processed for mitochondria purification based on established protocols (*Millar et al., 2007*). Briefly, hypocotyls were homogenized in a Waring blender with homogenization buffer (0.4 M sucrose, 1 mM EDTA, 25 mM MOPS-KOH, 10 mM tricine, 1% w:v PVP-40, freshly added 8 mM cysteine and 0.1% w:v BSA, pH 7.8) before a centrifugation of 10 min at 1000 x *g* (4 °C). The supernatant was collected and centrifuged for 30 min at 12,000 x *g* (4 °C). The resulting pellet was resuspended with wash buffer (0.4 M sucrose, 1 mM EDTA, 25 mM MOPS-KOH, freshly added 0.1% w:v BSA, pH 7.2) and gently centrifuged at 1000 x *g* for 5 min (4 °C). This supernatant was then centrifuged for 45 min at 12,000 x *g*. The resulting pellet was resuspended in wash buffer, loaded on to sucrose step gradients (35% w:v, 55% w:v, 75% w:v) and centrifuged for 60 min at 52,900 x *g*. The sucrose gradients were fractionated with a BioComp Piston Gradient Fractionator connected to a Gilson F203B fraction collector, following absorbance at 280 nm. The fractions containing mitochondria were pooled, diluted 1:5 in 10 mM MOPS-KOH, 1 mM EDTA, pH 7.2 and centrifuged for 20 min at 12,000 x *g* (4 °

C). The pellet was resuspended in final resuspension buffer (20 mM HEPES, 50 mM NaCl, 1 mM EDTA, 10% glycerol, pH 7.5) and centrifuged for 20 min at 16,000 x *g* (4 ℃). The supernatant was removed, and the pellets were frozen and stored in a −80 ℃ freezer. The yield of these mitochondrial pellets was 0.8–1 mg per gram of hypocotyl.

### *Vigna radiata* mitochondrial membrane wash

Frozen *V. radiata* mitochondrial pellets were thawed at 4℃, resuspended in 10 ml of chilled (4 ℃) double-distilled water per gram of pellet and homogenized with a cold Dounce glass homogenizer on ice. Chilled KCl was added to the homogenate to a final concentration of 0.15 M and further homogenized. The homogenate was centrifuged for 45 min at 32,000 x *g* (4 ℃). The pellets were resuspended in cold Buffer M (20 mM Tris, 50 mM NaCl, 1 mM EDTA, 2 mM DTT, 0.002% PMSF, 10% glycerol, pH 7.4) and further homogenized before centrifugation at 32,000 x *g* for 45 min (4 ℃). The pellets were resuspended in 3 ml of Buffer M per gram of starting material and further homogenized. The protein concentration of the homogenate was determined using a Pierce BCA assay kit (Thermo Fisher, Waltham, Massachusetts, USA), and the concentration was adjusted to a final concentration of 10 mg/ml and 30% glycerol.

### Extraction and purification of mitochondrial complexes

Washed membranes were thawed at 4℃. Digitonin (EMD Millipore, Burlington, Massachusetts, USA) was added to the membranes at a final concentration of 1% (w:v) and a digitonin:protein ratio of 4:1 (w:w). Membrane complexes were extracted by tumbling this mixture for 60 min at 4 ℃. The extract was centrifuged at 16,000 x *g* for 45 min (4 ℃). Amphipol A8-35 (Anatrace, Maumee, Ohio, USA) was added to the supernatant at a final concentration of 0.2% w:v and tumbled for 30 min at 4℃, after which gamma-cyclodextrin (EMD Millipore, Burlington, Massachusetts, USA) was added stepwise to a final amount of 1.2x gamma-cyclodextrain:digitonin (mole:mole). The mixture was centrifuged at 137,000 x *g* for 60 min (4 ℃). The supernatant was concentrated with centrifugal protein concentrators (Pall Corporation, NY, NY, USA) of 100,000 MW cut-off, loaded onto 10–45% (w:v) or 15–45% (w:v) linear sucrose gradients in 15 mM HEPES, 20 mM KCl, pH 7.8 produced using factory settings of a BioComp Instruments (Fredericton, Canada) gradient maker and centrifuged for 16 hr at 37,000 x *g* (4 ℃). The gradients were subsequently fractionated with a BioComp Piston Gradient Fractionator connected to a Gilson F203B fraction collector, following absorbance at 280 nm. For grid preparation, the relevant fractions were buffer-exchanged into 20 mM HEPES, 150 mM NaCl, 1 mM EDTA, pH 7.8 (no sucrose) and concentrated to a final protein concentration of 6 mg/ml and mixed one-to-one with the same buffer containing 0.2% digitonin (w:v), for a final concentration of 0.1% digitonin (w:v).

### NADH-dehydrogenase in-gel activity assay with blue-native polyacrylamide gel electrophoresis (BN-PAGE)

Mitochondrial membrane extractions were diluted in 2X BN-loading buffer (250 mM aminocaproic acid, 100 mM Tris-HCl, pH 7.4, 50% glycerol, 2.5% (w:v) Coomassie G-250), loaded on pre-cast 3–12% NativePAGE Bis-Tris gels (Invitrogen, Carlsbad, CA) and run at 4 ℃. The cathode buffer was 50 mM Tricine, 50 mM BisTris-HCl, pH 6.8 plus 1X NativePAGE Cathode Buffer Additive (0.02% Coomassie G-250) (Invitrogen, Carlsbad, CA) and the anode buffer was 50 mM Tricine, 50 mM BisTris-HCl, pH 6.8. Gels were run at 150 V constant voltage for ~30 min, after which the cathode buffer was switched for a 'light blue' cathode buffer containing 50 mM Tricine, 50 mM BisTris-HCl, pH 6.8 plus 0.1X NativePAGE Cathode Buffer Additive (0.002% Coomassie G-250) (Invitrogen, Carlsbad, CA). The settings were changed to 200 V constant voltage and run for another ~90 min.

The CI in-gel NADH-dehydrogenase activity assay was performed based on *Schertl and Braun, 2015*. The BN-PAGE gel was incubated in 10 ml of freshly prepared reaction buffer (1.5 mg/ml nitrotetrazoleum blue in 10 mM Tris-HCl pH 7.4). Freshly thawed NADH stock (20 mM) was added to the container with the gel, to a final concentration of 150 µM. The gel with the complete reaction buffer was rocked at room temperature for ~10 min. Once purple bands indicating NADH-dehydrogenase activity appeared, the reaction was quenched with a solution of 50% methanol (v:v) and 10% acetic acid (v:v).

## Complex III$_2$ spectroscopic activity assays

Spectroscopic activity assays were performed based on *Letts et al., 2019*; *Huang et al., 2015*; *Barrientos et al., 2009*, with modifications. Reduced-decylubiquinone (DQ): cyt *c* activity was measured by spectroscopic observation of cyt *c* reduction at 550 nm wavelength at room temperature using a Molecular Devices (San Jose, CA) Spectramax M2 spectrophotometer. Reactions were carried out in 96-well plates. DQ (Santa Cruz Biotechnology, Dallas, TX) was freshly reduced. The required amount of ethanol-diluted 100 mM DQ was aliquoted and further diluted to ~300 µl with 100% ethanol. A couple of lithium borohydride crystals were added to reduce the DQ, turning the solution transparent. Excess lithium borohydride was quenched by the dropwise addition of 1 N HCl until no further bubbles were observed. The ethanol was then evaporated with a stream of argon gas until a volume of ~50 µl was obtained. This reduced-DQ was added to a master mix reaction buffer (100 mM HEPES, pH 7.8, 50 mM NaCl, 10% glycerol, 0.1% CHAPS, 1 mg/ml BSA, 0.25 mg/ml 4:1 asolectin:cardiolipin in 0.1% CHAPS, 25 U/ml SOD, 4 µM KCN, 15 µM piericidin) at a final concentration of 100 µM and mixed by vortexing. The pH of the reaction buffer master mix was checked to be 7–8 with pH strips. The reaction master mix was aliquoted, and CIII$_2$ inhibitors antimycin A and myxothiazol were added at 1 µM final concentration where relevant and mixed by vortexing. Protein samples (5 µg) were added to the respective aliquots of reaction buffer to a total volume of 200 µl and mixed by vortexing. The reaction was initiated by addition of equine cyt *c* (Sigma Aldrich, St Louis, MO) to a final concentration of 100 µM, briefly mixed by pipetting and plate stirring for 10 s before recording for 3 min every 4 s. Measurements were done in 3–5 replicates, averaged and background-corrected. An extinction co-efficient of 28 mM$^{-1}$ cm$^{-1}$ (*Huang et al., 2015*) was used in the calculations. Statistical significance was determined using two-tailed t-tests.

## Complex IV spectroscopic activity assays

Spectroscopic activity assays were performed based on *Letts et al., 2019*; *Huang et al., 2015*; *Barrientos et al., 2009*, with modifications. Cytochrome *c* oxidase activity was measured by spectroscopic observation of the oxidation of reduced cyt *c* at 550 nm wavelength at room temperature using a Molecular Devices (San Jose, CA) Spectramax M2 spectrophotometer. Reactions were carried out in 96-well plates. Equine cyt *c* (Sigma Aldrich, St Louis, MO), diluted in 20 mM HEPES, pH 7.4, 50 mM NaCl, 10% glycerol buffer, was freshly reduced based on manufacturer's instructions with modifications. Dithiothreitol (DTT) was added at a 10 mM final concentration. After ~20 min and a visible change in color, cyt *c* reduction was confirmed spectroscopically. Given the spectrophotometer's specifications, the simultaneous measurement of A$_{550}$ and A$_{565}$ was suboptimal (A$_{550}$: A$_{565}$ ratio of ~9). Therefore, the A$_{550}$:A$_{575}$ was measured instead, as per manufacturer's recommendations. Cyt *c* reduction was confirmed at A$_{550}$:A$_{575}$ ratio ~22–24. For the spectroscopic activity assay, the reaction master mix consisted of 20 mM HEPES, pH 7.4, 50 mM NaCl, 10% glycerol, 0.1% CHAPS (w:v) with additional 4 µM KCN wherever appropriate. Protein samples (5 µg) were added to the respective aliquots of reaction buffer to a total volume of 200 µl and mixed by vortexing. The reaction was initiated by addition of reduced cyt *c* to a final concentration of 100 µM, briefly mixed by pipetting and plate stirring for 10 s before recording for 3 min every 4 s. Measurements were done in 3–4 replicates, averaged and background-corrected. An extinction co-efficient of 28 mM$^{-1}$ cm$^{-1}$(*Huang et al., 2015*) was used in the calculations. Statistical significance was determined using two-tailed t-tests.

## Mass spectrometry

The sample used for mass spectrometry was the sample used to blot the cryoEM grid that was used for here and in *Maldonado et al., 2020*. This sample corresponds to concentrated, pooled peak two fractions from the sucrose gradient shown in *Maldonado et al., 2020* (fractions 10–11, *Figure 1— figure supplement 2H*). This sample is roughly equivalent to fractions 11–13 from *Figure 1—figure supplement 4* here. Thus, the mass spectrometry results of this mixed sample include complex I subunits in addition to CIII$_2$ and CIV subunits. See below for full dataset availability and accession codes.

Samples were digested with the S-Trap micro (PROTIFI) digestion. Digestion followed the S-trap protocol. The proteins were reduced and alkylated, the buffer concentrations were adjusted to a final concentration of 5% SDS, 50 mM TEAB, 12% phosphoric acid was added at a 1:10 (v:v) ratio

with a final concentration of 1.2% and S-trap buffer (100 mM TEAB in 90% MEOH) was added at a 1:7 ratio (v:v) ratio. The protein lysate S-trap buffer mixture was then spun through the S-trap column and washed three times with S-Trap buffer. Finally, 50 mM TEAB and 1 µg of trypsin (1:25 ratio) was added and the sample was incubated overnight with one addition of 50 mM TEAB and trypsin after two hours. The following day the digested peptides were released from the S-trap solid support by spinning at 1 min for 3000 x g with a series of solutions starting with 50 mM TEAB which is placed on top of the digestion solution then 5% formic acid followed by 50% acetonitrile, 0.1% formic acid. The solution was then vacuum centrifuged to almost dryness and resuspended in 2% acetonitrile, 0.1% triflouroacetic acid (TFA) and subjected to Fluorescent Peptide Quantification (Pierce).

Digested peptides were analyzed by LC-MS/MS on a Thermo Scientific Q Exactive plus Orbitrap Mass spectrometer in conjunction Proxeon Easy-nLC II HPLC (Thermo Scientific) and Proxeon nano-spray source. The digested peptides were loaded on a 100 micron x 25 mm Dr. Masic reverse phase trap where they were desalted online before being separated using a 75 micron x 150 mm Magic C18 200 Å 3U reverse phase column. Peptides were eluted using a 70 min gradient with a flow rate of 300 nL/min. An MS survey scan was obtained for the m/z range 300–1600, MS/MS spectra were acquired using a top 15 method, where the top 15 ions in the MS spectra were subjected to HCD (High Energy Collisional Dissociation). An isolation mass window of 2.0 m/z was used for the precursor ion selection, and normalized collision energy of 27% was used for fragmentation. A twenty second duration was used for the dynamic exclusion.

Tandem mass spectra were extracted and charge state deconvoluted by Proteome Discoverer (Thermo Scientific). All MS/MS samples were analyzed using X! Tandem (The GPM, thegpm.org; version X! Tandem Alanine (2017.2.1.4)). X! Tandem was set up to search the Uniprot *Vigna radiata* database (October 2019, 35065 entries) the cRAP database of common laboratory contaminants (http://www.thegpm.org/crap; 117 entries) plus an equal number of reverse protein sequences assuming the digestion enzyme trypsin. X! Tandem was searched with a fragment ion mass tolerance of 20 PPM and a parent ion tolerance of 20 PPM. Carbamidomethyl of cysteine and selenocysteine was specified in X! Tandem as a fixed modification. Glu->pyro Glu of the n-terminus, ammonia-loss of the n-terminus, gln->pyro Glu of the n-terminus, deamidated of asparagine and glutamine, oxidation of methionine and tryptophan and dioxidation of methionine and tryptophan were specified in X! Tandem as variable modifications.

Scaffold (version Scaffold_4.8.4, Proteome Software Inc, Portland, OR) was used to validate MS/MS based peptide and protein identifications. Peptide identifications were accepted if they could be established at greater than 98.0% probability by the Scaffold Local FDR algorithm. X! Tandem identifications required score of at least 2. Protein identifications were accepted if they could be established at greater than 6.0% probability to achieve an FDR less than 5.0% and contained at least two identified peptides. Protein probabilities were assigned by the Protein Prophet algorithm (*Nesvizhskii et al., 2003*). Proteins that contained similar peptides and could not be differentiated based on MS/MS analysis alone were grouped to satisfy the principles of parsimony. Proteins sharing significant peptide evidence were grouped into clusters.

## CryoEM data acquisition

The sample (6 mg/ml protein in 20 mM HEPES, 150 mM NaCl, 1 mM EDTA, 0.1% digitonin, pH 7.8) was applied onto glow-discharged holey carbon grids (Quantifoil, 1.2/1.3 300 mesh) followed by a 60 s incubation and blotting for 9 s at 15℃ with 100% humidity and flash-freezing in liquid ethane using a FEI Vitrobot Mach III.

CryoEM data acquisition was performed on a 300 kV Titan Krios electron microscope equipped with an energy filter and a K3 detector at the UCSF W.M. Keck Foundation Advanced Microscopy Laboratory, accessed through the Bay Area CryoEM Consortium. Automated data collection was performed with the SerialEM package (*Schorb et al., 2019*). Micrographs were recorded in super-resolution mode at a nominal magnification of 60,010 X, resulting in a pixel size of 0.8332 $Å^2$. Defocus values varied from 1.5 to 3.0 µm. The dose rate was 20 electrons per pixel per second. Exposures of 3 s were dose-fractionated into 118 frames, leading to a dose of 0.72 electrons per $Å^2$ per frame and a total accumulated dose of 51 electrons per $Å^2$. A total of 9816 micrographs were collected.

## Data processing

Software used in the project was installed and configured by SBGrid (*Morin et al., 2013*). All processing steps were done using cryoSPARC and RELION 3.0 (*Zivanov et al., 2018*; *Punjani et al., 2017*) unless otherwise stated. Motioncor2 (*Zheng et al., 2017*) was used for whole-image drift correction of each micrograph. Contrast transfer function (CTF) parameters of the corrected micrographs were estimated using Ctffind4 (*Rohou and Grigorieff, 2015*). After motion correction and CTF correction, a set of 8541 micrographs was selected for further processing. Automated particle picking using crYOLO (*Wagner et al., 2019*; *Wagner and Raunser, 2020*) resulted in ~1.5 million particles. The particles were extracted using $400^2$ pixel box binned two-fold and sorted by reference-free 2D classification in Relion using (–max_sig 5), followed by re-extraction at $512^2$ pixel box. Reference-free 2D classification in Relion resulted in the identification of 502,224 particles that were then imported into cryoSPARC for further reference-free 2D classification. A set of 121,702 particles were identified by 2D classification in cryoSPARC to contain $CIII_2$ alone or SC $III_2$+IV (*Figure 1—figure supplement 1*). These particles were subjected to *ab initio* model generation with four targets to remove contaminant particles resulting in a set of 99,937 particles across three classes. Each individual class was subjected to an additional round of *ab initio* model generation with three targets. This separated $CIII_2$ alone particles from the SC $III_2$+IV class and allowed the recovery of $CIII_2$ alone particles from the poor particle class. $CIII_2$ alone particles from across the *ab initio* model generation jobs were pooled, defining a final class of 48,111 particles. The multiple rounds of *ab initio* model generation resulted in only one good class of 28,020 SC $III_2$+IV particles. Poses for these two particle sets ($CIII_2$ alone and SC $III_2$+IV) were refined using cryoSPARC's Homogeneous Refinement (New) algorithm including Defocus Refinement and Global CTF refinement. This resulted in reconstructions at 3.2 Å and 3.8 Å resolution for $CIII_2$ alone and SC $III_2$+IV respectively, according to the gold standard FSC criteria (*Figure 1—figure supplement 1*; *Scheres and Chen, 2012*).

In parallel, a set of 69,876 particles were identified by further 2D and 3D classification in Relion (*Figure 1—figure supplement 2*). These particles, which contained a mixture of SC $III_2$+IV and $CIII_2$ alone particles, were aligned using a SC $III_2$+IV model and mask. They were then subjected to five rounds of masked classification using a CIV model and mask aligned with the position of CIV in the SC. Three parallel masked classifications (all using T = 8) varied in the degree of rotational searches, with either no searches, 0.1° sampling interval over +/- 0.2° search range and 3.7° sampling interval over +/- 7.5° search range. The masked 3D classification without searches was repeated successively for three rounds, inputting the best particles from the previous round into the subsequent round. The best CIV class from each 3D classification were selected and combined while removing overlaps (any particle within 200 pixels of another was considered as an overlap and discarded). This 3D classification strategy resulted in a set of 38,410 particles. The coordinates of these particles were used to extract two sets of re-centered SC particles, one centered on $CIII_2$ and one centered on CIV. These two sets of particles were independently 3D-refined, CTF-refined and Bayesian-polished using a model and mask centered around $CIII_2$ or CIV respectively. The CIV-centered shiny particles were subjected to a final round of 3D classification, defining a final set of 29,348 CIV particles. Although this final round of 3D classification did not improve the nominal resolution of the map, it increased map quality at the periphery of the complex. These final $CIII_2$ and CIV classes resulted in reconstructions at 3.7 Å and 3.8 Å resolution for $CIII_2$ and IV from the SC respectively, according to the gold standard FSC criteria (*Figure 1—figure supplement 1*; *Scheres and Chen, 2012*). These two maps were aligned with the full SC map and combined to make a composite map using Phenix.

3D variability analysis (3DVA) was performed on $CIII_2$ alone in cryoSPARC using their built-in algorithm (*Punjani and Fleet, 2020*). Two separate instances of 3DVA were performed, each solving for the four largest principal components. The first instance used a mask around the entire $CIII_2$ and data was low-pass filtered to 6 Å resolution to remove the influence of high-resolution noise from the amphipol detergent belt. The second instance used a mask focused around the IMS domain of $CIII_2$ and was low-pass filtered to 5 Å resolution. All other parameters were kept as default.

## Model building and refinement

Starting template models for *V. radiata* $CIII_2$ was ovine $CIII_2$ (PDB: 6Q9E). Starting template models for *V. radiata* CIV were from *S. cerevisiae* (PDB: 6HU9) and bovine (PDB: 5B1A) CIV. Additionally, starting models for the *V. radiata* subunits were generated using the Phyre2 web portal

(*Kelley et al., 2015*). Real-space refinement of the model was done in PHENIX (*Liebschner et al., 2019*; *Goddard et al., 2018*; *Pettersen et al., 2004*) and group atomic displacement parameters (ADPs) were refined in reciprocal space. The single cycle of group ADP refinement was followed by three cycles of global minimization, followed by an additional cycle of group ADP refinement and finally three cycles of global minimization (*Letts et al., 2019*). The refined CIII$_2$ and CIV models were docked into the SC III$_2$+CIV map without subsequent refinement.

## Model interpretation and figure preparation

Molecular graphics and analyses were performed with UCSF Chimera (*Pettersen et al., 2004*) and ChimeraX (*Goddard et al., 2018*) developed by the Resource for Biocomputing, Visualization, and Informatics at the University of California, San Francisco, with support from NIH P41-GM103311 and R01-GM129325 and the Office of Cyber Infrastructure and Computational Biology, National Institute of Allergy and Infectious Diseases. PyMOL Molecular Graphics System, Version 2.0 Schrödinger, LLC was also used.

## Acknowledgements

We are grateful to K Abe, MG Zaragoza, C Goodwin, R Murguia and C Bower for help with *V. radiata* growth and mitochondrial isolations. Data was collected at the UCSF WM Keck Foundation Advanced Microscopy Laboratory, accessed through the Bay Area CryoEM Consortium BACEM, with the assistance of D Bulkley and Z Yu. We are grateful to W Broadly of the UC Davis High-Performance Cluster and M Salemi and B Phinney of the UC Davis Proteomics Core for technical assistance. MM acknowledges funding from the UC Davis POP Program.

## Additional information

### Funding
No external funding was received for this work.

### Author contributions
Maria Maldonado, Conceptualization, Resources, Data curation, Formal analysis, Supervision, Funding acquisition, Validation, Investigation, Visualization, Methodology, Writing - original draft, Project administration, Writing - review and editing; Fei Guo, Formal analysis, Investigation; James A Letts, Conceptualization, Resources, Data curation, Formal analysis, Supervision, Funding acquisition, Validation, Visualization, Methodology, Project administration, Writing - review and editing

### Author ORCIDs
Maria Maldonado (iD) https://orcid.org/0000-0002-3428-1053
James A Letts (iD) https://orcid.org/0000-0002-9864-3586

### Decision letter and Author response
Decision letter https://doi.org/10.7554/eLife.62047.sa1
Author response https://doi.org/10.7554/eLife.62047.sa2

## Additional files

### Supplementary files
• Supplementary file 1. (**a**) Mass spectrometry identification of *V. radiata* CIII$_2$ and CIV subunits. (**b**) Model-building statistics by subunit. (**c**) CIII$_2$ and CIV subunit homologs in plants, yeast and mammals. *V. radiata* homologs were obtained by performing BLASTp searches of the *Arabidopsis thaliana* genes (*Meyer et al., 2019*). Mammalian and yeast homologs were obtained from *Hartley et al., 2019*; *Maréchal et al., 2012*. Additional BLASTp searches were performed as needed. (**d**) RNA edits identified in *V. radiata* CIII$_2$ and CIV subunits. RNA-editing sites were initially identified by an unambiguous mismatch in the cryoEM density and the expected density for the mitochondrial-DNA-encoded residue. The existence of the RNA-editing site in other plants, or the implied restoration of

the consensus sequence was then inspected. Amino-acid changes to the atomic model (with respect to the mt-DNA sequence) were only made for amino-acid positions that had unambiguous cryoEM density evidence and whose editing is conserved or would restore the conserved sequence.

- Transparent reporting form

## Data availability

PDB accession codes: 7JRG, 7JRO, 7JRP. EMBD accession codes: 22445, 22449, 22450, 22447, 22448. EMPIAR accession code: EMPIAR-10586.

The following datasets were generated:

| Author(s) | Year | Dataset title | Dataset URL | Database and Identifier |
|---|---|---|---|---|
| Maldonado M, Guo F, Letts JA | 2020 | Mass spectrometry data for Vigna radiata concentrated, pooled peak 2 fractions from sucrose gradient, Maldonado et al 2020 | https://massive.ucsd.edu/ProteoSAFe/dataset.jsp?task=22e3e0e4-b0714a008d882918de-da9888 | MassIVE, MSV0000 86237 |
| Maldonado M, Guo F, Letts JA | 2020 | Mass spectrometry data for Vigna radiata concentrated, pooled peak 2 fractions from sucrose gradient, Maldonado et al 2020 | http://proteomecentral.proteomexchange.org/cgi/GetDataset?ID=PXD021850 | ProteomeXchange, PXD021850 |
| Maldonado M, Letts JA | 2020 | Plant Mitochondrial complex III2 from Vigna radiata | https://www.rcsb.org/structure/7JRG | RCSB Protein Data Bank, 7JRG |
| Maldonado M, Letts JA | 2020 | Plant Mitochondrial complex IV from Vigna radiata | https://www.rcsb.org/structure/7JRO | RCSB Protein Data Bank, 7JRO |
| Maldonado M, Letts JA | 2020 | Plant Mitochondrial complex SC III2 +IV from Vigna radiata | https://www.rcsb.org/structure/7JRP | RCSB Protein Data Bank, 7JRP |
| Maldonado M, Letts JA | 2020 | Plant Mitochondrial complex III2 from Vigna radiata | http://emsearch.rutgers.edu/atlas/22445_summary.html | EMDataBank, 22445 |
| Maldonado M, Letts JA | 2020 | Plant Mitochondrial complex IV from Vigna radiata | http://emsearch.rutgers.edu/atlas/22449_summary.html | EMDataBank, 22449 |
| Maldonado M, Letts JA | 2020 | Plant Mitochondrial complex SC III2 +IV from Vigna radiata composite map | http://emsearch.rutgers.edu/atlas/22450_summary.html | EMDataBank, 22450 |
| Maldonado M, Letts JA | 2020 | Plant mitochondrial supercomplex III2+IV from Vigna radiata | http://emsearch.rutgers.edu/atlas/22447_summary.html | EMDataBank, 22447 |
| Maldonado M, Letts JA | 2020 | Plant mitochondrial complex III2 from Vigna radiata supercomplex III2+IV, focused refinement | http://emsearch.rutgers.edu/atlas/22448_summary.html | EMDataBank, 22448 |
| Maldonado M, Padavannil A, Zhou L, Guo F, Letts JA | 2020 | Cryo electron micrographs of digitonin-solubilized, amphipol-stabilized, sucrose-gradient-purified V. radiata mitochondrial membranes - mixed fraction containing CI*, CIII2 and SC III2+IV | https://www.ebi.ac.uk/pdbe/emdb/empiar/entry/10586/ | Electron Microscopy Public Image Archive, EMPIAR-10586 |

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
