## [Decision Letter]

**Acceptance summary:**

Your paper opens reports the first structures of respiratory chain components, Complex III (CIII2) and IV (CIV), from plants. There are a number of unique features of these complexes not found in mammalian or fungal systems. Your work forms a fundamental addition to our knowledge of respiratory complexes and generates hypotheses that will be further tested by the field for years to come.

**Decision letter after peer review:**

Thank you for submitting your article "Atomic structures of respiratory complex III2 , complex IV and supercomplex III2+IV from vascular plants" for consideration by *eLife*. Your article has been reviewed by three peer reviewers, one of whom is a member of our Board of Reviewing Editors, and the evaluation has been overseen by Kenton Swartz as the Senior Editor. The following individuals involved in review of your submission have agreed to reveal their identity: Hans-Peter Braun (Reviewer #2); John L Rubinstein (Reviewer #3).

The reviewers have discussed the reviews with one another and the Reviewing Editor has drafted this decision to help you prepare a revised submission.

Summary

Mitochondrial ATP synthesis requires a proton gradient across the mitochondrial membranes. This gradient is established by the Respiratory Complexes, transfer electrons from electron donors NADH to Quinones and couple this to the pumping of protons across the membrane. While more insight into yeast and mammalian respiratory complexes has become available to understand this process, the high-resolution structural details of the respiratory complexes and super-complexes of plants have remained mostly unknown. Respiratory complex III (CIII2) in plants has an additional mitochondrial processing peptidase activity which is carried out by separate complexes in yeast and mammals. CIII2 forms a supercomplex with Complex IV (CIV) and structures of the separate and supercomplex were unknown.

In this manuscript, the authors report the structure of respiratory complexes III, IV, and a III2IV supercomplex from vascular plants. Although there is structural information on supercomplexes from several organisms, the structures of plant supercomplexes remain unknown, defining an exciting research frontier.

Major comments

1) The authors perform NADH:quinone oxidoreductase activity of the complex, however, they provide no evidence of amphipol stabilized CIII2 respiratory activity, MPP domain peptidase activity, CIV activity, or supercomplex activity. Ideally, the manuscript would include experimental evidence of these activities. This evidence would allow the authors to more confidently discuss how the structural features and dynamics relate to enzymatic properties. In the absence of this evidence of activity in their preparation the authors should be sure to emphasize throughout the manuscript, where appropriate, that they do not know if their preparation is active.

2) The authors find interesting dynamics using the 3DVA algorithm in cryoSPARC and present these in Figure 4 of the manuscript and several videos. The dynamics presented in panel C of the figure are easy to follow; however, it is difficult to see exactly what it happening in panel A and B. Although the videos make these motions the figures could be improved. For example, figures in the original 3DVA paper (Punjani and Fleet, 2020) clearly show the structural changes present in their systems.

Interestingly, the authors video from 3D variability ISP of one CIII monomer is in the b position the other is in the c1 position, and vice versa. This situation is analogous to the movement of the cyt. cc domain of the *M. smegmatis* CIII2CIV2 supercomplex (Wiseman et al., 2018), which moves instead of the ISP in that system. Anticorrelated ISP movement would have mechanistic implications for the CIII2 dimer. In the *M. smegmatis* study a single asymmetric structure was determined with one cc domain up and one cc domain down, which was used to infer the anticorrelated movement. In the present structure the anticorrelated movement is detected from a single PCA vector. The authors do not state explicitly that their data show anticorrelated movement of the ISP but the video suggests it strongly. It seems likely that if the movement of the ISPs were anticorrelated, the authors would have isolated a single asymmetric structure. Correlated movement of the ISPs may be detected in a different 3D PCA vector. Therefore, unless the authors have strong evidence for anticorrelated ISP position they should ensure that their manuscript does not imply that it occurs.

3) Starting material for structure determination is a fraction containing a complex I intermediate. Particles representing complexes III2, IV and supercomplex III+IV were purified in silico. This should be illustrated in Figure 1—figure supplement 1. The micrograph shown in part A of the figure should mainly include the complex I intermediate. Throughout the supplement, the numbers of particles should be given for the complex I intermediate, III2, IV and supercomplex III+IV.

4) The purification of III2 and IV is "supported by mass spectrometry data (Table 1)". However, no primary MS data are shown and no details on the experiments. I assume that MS has been carried out with the mixed mitochondrial fraction including the complex I intermediate and all other complexes present in this fraction, including subunits of complex III and IV. Presumably, several hundreds of proteins were identified in this fraction. This experiment certainly does not allow to systematically identify subunits present in the protein complexes of interest. The complete MS datasets should be presented.

5) The authors certainly are experts in single particle cryo electron microscopy, but new in the field of plant biology. The authors are not aware of several key publications on the respiratory chain of plants. Citations have to be substantially updated.

---

## [Author Response]

Major comments1) The authors perform NADH:quinone oxidoreductase activity of the complex, however, they provide no evidence of amphipol stabilized CIII2 respiratory activity, MPP domain peptidase activity, CIV activity, or supercomplex activity. Ideally, the manuscript would include experimental evidence of these activities. This evidence would allow the authors to more confidently discuss how the structural features and dynamics relate to enzymatic properties. In the absence of this evidence of activity in their preparation the authors should be sure to emphasize throughout the manuscript, where appropriate, that they do not know if their preparation is active.

We have performed activity assays for respiratory activities of CIII_2_ and CIV from a preparation equivalent to that used for the cryoEM sample and show the results in the new Figure 1—figure supplement 4. CIII_2_ and CIV are active and inhibitable by established inhibitors (antimycin A and myxothiazol for CIII_2_, potassium cyanide for CIV) as shown by spectroscopic activity assays (Huang et al., 2015).

We also performed MPP peptidase activity assays based on Teixeira et al., 2015 with the complete pre-sequence of the F_1_b subunit of the *Nicotiana plumbaginifolia* ATP synthase as previously reported (Braun et al., 1992). Although we see peptidase activity from the washed mitochondrial membranes from which the complexes were extracted, our peptidase assays from the purified fractions did not show clear activity. We believe this may be due to the early use of EDTA in the purification and freezing buffers, as well as to unoptimized conditions for the peptidase activity. Given the pandemic-related limitations on the number of people who can be simultaneously present in the lab and working together in teams, we are not at this time able to produce enough mitochondrial samples from etiolated mung beans for further troubleshooting and optimization of the incubation and pre-incubation conditions for this assay. Nevertheless, the activity in the washed membranes and excellent superposition of the MPP domains of our structure with the previously published structure of soluble, active MPP from yeast (as discussed in the manuscript), strongly suggest that MPP should also be active in our preparation although we are not able to definitively show it at this point. We do not believe this changes our interpretation of the structures but do emphasize our lack of confirmation in the Results.

New text: “Further examination using spectroscopic activity assays^[52]^ confirmed that pooled fractions of the preparation contained CIII_2_ respiratory activity (from reduced decylubiquinone to cytochrome *c*) that was inhibited by CIII_2_ inhibitors antimycin A and myxothiazol (Figure 1—figure supplement 4). The sample also showed CIV activity from reduced cytochrome *c* to oxygen that was inhibited by CIV inhibitor potassium cyanide (Figure 1—figure supplement 4). Although mitochondrial processing peptidase (MPP) activity of CIII_2_ was recovered from isolated *V. radiata* mitochondrial membranes (not shown), MPP activity assays^[22, 53]^ of the pooled fractions were inconclusive. Owing to research restrictions during the 2020 COVID-19 pandemic, we were not able to further optimize the peptidase assay for the pooled fractions. However, given the high superposition between the *V. radiata* MPP domain shown here and structures of active MPP, we do not believe our inability to confirm MPP activity in this case significantly impacts our interpretation of the data.”

2) The authors find interesting dynamics using the 3DVA algorithm in cryoSPARC and present these in Figure 4 of the manuscript and several videos. The dynamics presented in panel C of the figure are easy to follow; however, it is difficult to see exactly what it happening in panel A and B. Although the videos make these motions the figures could be improved. For example, figures in the original 3DVA paper (Punjani and Fleet, 2020) clearly show the structural changes present in their systems.

Thank you for the suggestion. Indeed, we found it challenging to capture the movement of the videos in panels A and B. We have updated these panels based on Punjani and Fleet’s figures.

Interestingly, the authors video from 3D variability ISP of one CIII monomer is in the b position the other is in the c1 position, and vice versa. This situation is analogous to the movement of the cyt. cc domain of the M. smegmatis CIII2CIV2 supercomplex (Wiseman et al., 2018), which moves instead of the ISP in that system. Anticorrelated ISP movement would have mechanistic implications for the CIII2 dimer. In the M. smegmatis study a single asymmetric structure was determined with one cc domain up and one cc domain down, which was used to infer the anticorrelated movement. In the present structure the anticorrelated movement is detected from a single PCA vector. The authors do not state explicitly that their data show anticorrelated movement of the ISP but the video suggests it strongly. It seems likely that if the movement of the ISPs were anticorrelated, the authors would have isolated a single asymmetric structure. Correlated movement of the ISPs may be detected in a different 3D PCA vector. Therefore, unless the authors have strong evidence for anticorrelated ISP position they should ensure that their manuscript does not imply that it occurs.

We agree with the reviewer that if the motion were completely anti-correlated, this should result in a single asymmetric structure. However, this is not what we observed and we have made this more clear in the manuscript. Further PCA vectors suggest that in some particles the position of the ISP may also be correlated, but the movement is less clear. In these PCA vectors the ISPs go from being ordered together in one position and therefore visible, to being completely disordered and thereby disappearing making it difficult to interpret. Only in the first PCA vector can the ISPs’ “motion” be tracked. We take the reviewer’s point that we should not give the impression of there being only anti-correlated movement without further comment. We edited/added sentences to clarify this issue:

Edited/new sentence: “Moreover, the motions of the UCR1 head domains of the CIII dimer in this variability component were anti-parallel, i.e. when one domain is in the proximal position, the other one is in the distal position and vice versa (Figure 4C, Video 10). However, weaker variability components also suggested that the position of UCR1 head domains may also be equivalent in some instances.”

New sentence: “Further studies are needed to examine to what extent the movement of the UCR1 head domains are correlated across the CIII dimer and what the mechanistic implications of parallel or anti-parallel movements may be.”

3) Starting material for structure determination is a fraction containing a complex I intermediate. Particles representing complexes III2, IV and supercomplex III+IV were purified in silico. This should be illustrated in Figure 1—figure supplement 1. The micrograph shown in part A of the figure should mainly include the complex I intermediate. Throughout the supplement, the numbers of particles should be given for the complex I intermediate, III2, IV and supercomplex III+IV.

We have updated the cryoEM processing workflow figure (Figure 1—figure supplement 1) with CI* 2D classes, particle numbers and the initial 3D classes. The figure contains a clear note referring the reader to our CI* paper for further details.

4) The purification of III2 and IV is "supported by mass spectrometry data (Table 1)". However, no primary MS data are shown and no details on the experiments. I assume that MS has been carried out with the mixed mitochondrial fraction including the complex I intermediate and all other complexes present in this fraction, including subunits of complex III and IV. Presumably, several hundreds of proteins were identified in this fraction. This experiment certainly does not allow to systematically identify subunits present in the protein complexes of interest. The complete MS datasets should be presented.

The reviewer is correct in his/her assumptions. We agree that the experiment does not allow us to systematically identify subunits present. We simply meant that the subunits were identified within the mixed MS sample. We have added a clarification in the main text and in the mass spec method section and have deposited the full MS dataset on the Massive and ProteomeExchange databases. We have added the accession numbers to our Materials and methods section. We welcome further suggestions as to how to present the MS data.

New sentence: “Complex III and CIV subunits were identified in the mixed mitochondrial pooled fraction by mass spectrometry, in addition to CI subunits (Supplementary file 1, see Materials and methods for full dataset availability).”

Materials and methods, Mass spectrometry: “The sample used for mass spectrometry was the sample used to blot the cryoEM grid that was used for here and in Maldonado et al., 2020. This sample corresponds to concentrated, pooled peak 2 fractions from the sucrose gradient shown in Maldonado et al., 2020 (fractions 10-11, Figure 1—figure supplement 2H). This sample is roughly equivalent to fractions 11-13 from Figure 1—figure supplement 4 here. Thus, the mass spectrometry results of this mixed sample include complex I subunits in addition to CIII2 and CIV subunits. See below for full dataset availability and accession codes.”

Materials and methods, Data availability: “Raw mass spectrometry files and search results are available from Massive (https://massive.ucsd.edu/) #MSV000086237 and ProteomeExchange (http://www.proteomexchange.org/) #PXD021850”

5) The authors certainly are experts in single particle cryo electron microscopy, but new in the field of plant biology. The authors are not aware of several key publications on the respiratory chain of plants. Citations have to be substantially updated.

We apologize for the unintended omissions and thank Dr Braun for the notes and corrections to ensure previous work is properly credited. We have updated our citations accordingly.